# Metabolic consequences of various fruit-based diets in a generalist insect species

**Laure Olazcuaga[1,2†], Raymonde Baltenweck[3†], Nicolas Leménager[1], Alessandra Maia-Grondard[3], Patricia Claudel[3], Philippe Hugueney[3‡], Julien Foucaud[1*‡]**

[1]UMR CBGP (INRAE-IRD-CIRAD, Montpellier SupAgro), Campus International de Baillarguet, Montferrier, France; [2]Department of Agricultural Biology, Colorado State University, Fort Collins, United States; [3]Université de Strasbourg, INRAE, Colmar, France

**Abstract** Most phytophagous insect species exhibit a limited diet breadth and specialize on a few or a single host plant. In contrast, some species display a remarkably large diet breadth, with host plants spanning several families and many species. It is unclear, however, whether this phylogenetic generalism is supported by a generic metabolic use of common host chemical compounds ('metabolic generalism') or alternatively by distinct uses of diet-specific compounds ('multi-host metabolic specialism')? Here, we simultaneously investigated the metabolomes of fruit diets and of individuals of a generalist phytophagous species, *Drosophila suzukii*, that developed on them. The direct comparison of metabolomes of diets and consumers enabled us to disentangle the metabolic fate of common and rarer dietary compounds. We showed that the consumption of biochemically dissimilar diets resulted in a canalized, generic response from generalist individuals, consistent with the metabolic generalism hypothesis. We also showed that many diet-specific metabolites, such as those related to the particular color, odor, or taste of diets, were not metabolized, and rather accumulated in consumer individuals, even when probably detrimental to fitness. As a result, while individuals were mostly similar across diets, the detection of their particular diet was straightforward. Our study thus supports the view that dietary generalism may emerge from a passive, opportunistic use of various resources, contrary to more widespread views of an active role of adaptation in this process. Such a passive stance towards dietary chemicals, probably costly in the short term, might favor the later evolution of new diet specializations.

**\*For correspondence:**
julien.foucaud@inrae.fr

[†]These authors contributed equally to this work
[‡]These authors also contributed equally to this work

**Competing interest:** The authors declare that no competing interests exist.

## Editor's evaluation

Most insect herbivores specialize in one or a few plants, but some species manage to be generalists and consume a wide variety of plant species. How generalists are able to cope with a variety of bioactive plant metabolites is not well understood. This important study uses untargeted metabolomics to identify patterns of metabolic generalism in herbivores. The study presents a convincing global chemical comparison of metabolites in insect diets with metabolites found in the insects, supporting inferences about the mechanisms underlying dietary generalism.

## Introduction

One of the most salient features of living organisms is their diversity in terms of ecological requirements. Some species show extreme specialization in their environment (e.g. obligate symbionts), while others are able to use a wide range of resources and tolerate many environmental conditions. Furthermore, these relationships between organisms and their biotic and abiotic environments are

**eLife digest** Most insects that feed on green plants are specialists, meaning that they feed on just a narrow range of plant species. This reduces competition, especially if the host plant contains chemical deterrents that are toxic to other insects. But specialists cannot easily switch to feed on other plants, making them vulnerable to changes in the availability of the particular food type that they eat.

Generalist insects, on the other hand, are able to consume a wide range of diets. This makes them more robust to changes in food availability, but it is unclear how these insects deal with the wider range of chemical compositions of their food. Do they convert food into energy using the same chemical process, or metabolism, for all the different things they eat? Or do generalists have a specific metabolic pathway for each food type?

To answer this question, Olazcuaga, Baltenweck et al. studied the metabolism of a generalist fruit fly species. The team compared four types of fruit (blackcurrant, cherry, cranberry and strawberry) and isolated separate groups of flies so that they each ate only one type of fruit. By comparing the chemical composition of the flies with that of the fruit they ate, they were able to work out how each fruit type was metabolised. They found that the flies converted food into energy using the same process regardless of the type of fruit they ate. This lack of a specialist metabolic pathway for each fruit type meant that some chemicals were not metabolised and accumulated in the fly's body instead. This build-up of unprocessed chemicals is likely to be harmful to the fly.

The results of Olazcuaga, Baltenweck et al. suggest that generalist insects do not actively adapt their metabolism to new food types. It's more likely that they try different types of food as the opportunity arises, regardless of the fact that some of the food will not be converted into energy and may harm them long term. These findings are important because they give us an insight into how the chemistry of a plant can shape the physiology of the organisms that consume it, and vice-versa. These insights are a crucial step in developing sustainable agriculture practices that must consider tackle how plants are pollinated, how plant seeds are dispersed and what type of pest control to use.

dynamic at various spatial and temporal scales. As such, the concepts of niche breadth and ecological specialization are of paramount importance in ecology and evolutionary biology (*Futuyma and Moreno, 1988*; *Sexton et al., 2017*).

The dietary requirements of phytophagous insects are a textbook example of variance in niche breadth (*Forister et al., 2012*; *Futuyma and Moreno, 1988*; *Jaenike, 1990*). In this group, the vast majority of species are specialists of one family of plants (*Forister et al., 2015*) and this specialization is often accompanied by extreme morphological and/or physiological specializations to exploit a particular plant (e.g. *Berenbaum, 1981*; *Hotti and Rischer, 2017*). In contrast, some rarer species display a much wider niche breadth, allowing them to consume up to several dozen plant families (*Clarke et al., 2005*; *Forister et al., 2015*). The larger proportion of specialists than generalist species (*Forister et al., 2015*), as well as the fact that transitions between specialism and generalism, are bidirectional (*Day et al., 2016*; *Janz et al., 2001*; *Nosil, 2002*; *Nylin et al., 2014*), raises questions about the origin and maintenance of generalism (*Forister et al., 2012*; *Singer, 2008*), and its biochemical basis.

From a physiological point of view, dietary generalism has been much less experimentally studied than specialism, probably in part because it requires the simultaneous consideration of numerous host species and the tracking of physiological responses that may involve far more components than specialism. Additionally, the study of dietary generalism lacks the intellectual appeal of bi-species interactions leading to biochemical and genetic arm-races. Indeed, many research scrutinized the evolution of plant defenses and their corresponding detoxifying pathways in herbivores (e.g. the glucosinolate–myrosinase system of Brassicaceae countered by *Spodoptera frugiperda*; *Winde and Wittstock, 2011*). Comparatively, knowledge about how generalist species cope with a wide variety of plant biochemistry is scarce (but see *Roy et al., 2016*).

The core question of dietary generalism at the level of the organism is how a generalist species metabolizes different diets, which involves some basic issues that are still pending. For instance, do individuals of a generalist species have a unique biochemical response to all diets or do they differ according to their diet? In evolutionary terms, do dietary generalist species exhibit a canalized

response to different dietary environments through a single, neutral, and generic use of hosts (H1; hereafter termed '*metabolic generalism*'), or do they show multiple specialized, adapted uses of different hosts (H2; hereafter termed '*multi-host metabolic specialism*')? Broad organismal-level patterns of biochemical composition can help answer this question because individuals on different diets should be chemically similar across diets according to the former hypothesis, or mostly dissimilar according to the latter (*Figure 1*).

To refine our understanding of diet generalism, we here use an untargeted metabolomics approach to characterize the metabolic impact of various fruit-based diets on the generalist insect species *Drosophila suzukii* (*Figure 1A*). The range of molecular compounds detected and quantified by current metabolomic methods may span many hundreds or thousands of molecules, ensuring a high-quality snapshot of the individuals' physiological state (*Johnson et al., 2016*). Additionally, untargeted metabolomics provides the opportunity to directly compare diets and individuals that consume them, in an objective and repeatable way. The invasive *Drosophila suzukii* (the spotted wing *Drosophila*) displays an exceptional diet breadth of over 20 plant families (*Asplen et al., 2015*; *Bellamy et al., 2013*; *Cini et al., 2014*; *Kanzawa, 1939*; *Kenis et al., 2016*; *Lee et al., 2011*; *Lee et al., 2015*; *Poyet et al., 2015*). While multiple experimental studies have found that different diets have an influence on its larval development (*Kenis et al., 2016*; *Olazcuaga et al., 2019*; *Poyet et al., 2015*), it is widely accepted that *D. suzukii* is a diet generalist species.

By directly comparing the metabolomes of fruits and flies that consume them, we investigated the following question: Do generalist individuals have a single physiological response to various diets ('*metabolic generalism*') or do they exhibit specific responses to these various diets ('*multi-host metabolic specialism*')? We first verified how biochemically different were the selected major host plant species used by *D. suzukii*. Then, we examined whether generalist individuals use different host plants in a similar way, qualitatively and quantitatively. Finally, we attempted to discriminate individuals of a generalist species according to their diet and identify diet-specific metabolites in generalist individuals.

## Results

The metabolomes of four major crop fruits (blackcurrant, cherry, cranberry, and strawberry) and their corresponding flies (that consumed them both at the larval and early adult stages) were characterized simultaneously using ultra-high-performance liquid chromatography coupled to mass spectrometry (UHPLC-MS; *Figure 1A*). Our untargeted approach enabled the detection and relative quantification of a set of 11,470 unique ions that were found in at least one fruit or fly sample. While the simultaneous processing of fruit and flies' samples provided a single *integral* dataset enabling direct comparison between diets and consumers, some analyses required the fruit-related and fly-related data to be separated, resulting in a *fruit* dataset and a *fly* dataset. Although the *integral*, *fruit*, and *fly* datasets are based on ions characterized by UHPLC-MS, some ions will be referred to as 'compounds' or 'metabolites' later in the manuscript, depending on the context. Importantly, we took care to standardize the *fly* and *integral* dataset using control flies (i.e. flies that developed on an artificial diet only) to avoid including ions derived from the artificial component of our fruit diets (see Methods section for more details). Controlled and raw *fly* datasets yielded similar results (see Appendix 1). We here present the results of the analyses of the controlled *fly* dataset.

### Metabolomic analysis of fruit composition

A preliminary requirement to investigate dietary generalism is to directly compare diets from a biochemical perspective, because different plant families may share chemical properties that could aid the transition from specialism to generalism (e.g. share the same plant defense compounds; *Bowers, 1983*). To exclude such misleading generalism, an often overlooked question is therefore whether the taxonomic scale (e.g. species) fairly represents different dietary challenges. We evaluated the biochemical proximity of the four investigated fruits using a Principal Component Analysis (PCA) framework on the *fruit* dataset. This analysis showed that the four investigated fruits were dissimilar and that technical replicates were highly similar (*Figure 1—figure supplement 1*). Blackcurrant was very dissimilar to all other fruit (discriminated on the first dimension of the PCA explaining 35% of the *fruit* dataset variance). Cherry was then discriminated by the second dimension of the PCA (28%

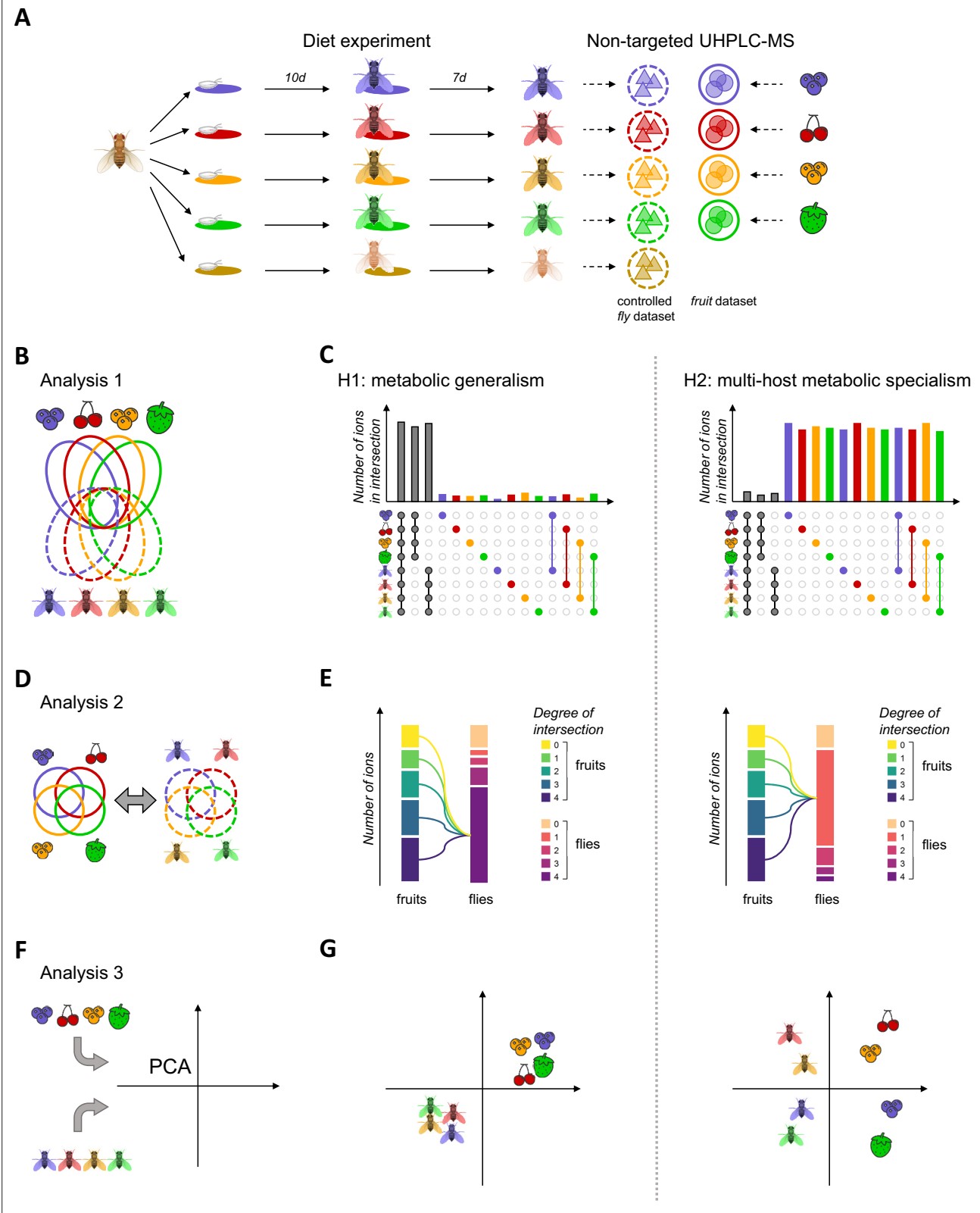

**Figure 1.** Schematic overview of experimental design, host use analyses, and expectations according to the '*metabolic generalism*' and '*multi-host metabolic specialism*' hypotheses. In order to investigate whether different food sources were treated similarly or differently by a generalist species, we followed a simple experimental procedure (**A**), where eggs of a single founder population were separately developed on four fruit media and one control 'German Food' medium for 10 days and kept on the same fruit for seven days of adult life. Both adult flies and fruit purees were processed

*Figure 1 continued*

to obtain the controlled *fly* and *fruit* datasets, with respective ions represented by colored triangles and dots. These datasets were used to perform three analyses (**B, D, F**) linked to three diverging expectations under our competing hypotheses of '*metabolic generalism*' and '*multi-host metabolic specialism*' (**C, E, G**). First, the comparison of the relative sizes of intersections of the complete dataset (**B**) should indicate whether ions are mostly shared between intersections of large sizes such as all fruits, all flies, or all samples (C, left) or whether ions are mostly confined to intersections of small size such as discrete categories or fruit-fly pairs (C, right). Second, relationships between intersections between fruits on one hand and between flies on the other (**D**) should indicate whether common or rare fruit ions are commonly (E, left) or rarely (E, right) found in flies. Third, while a principal component analysis (PCA) of the complete dataset should separate fruit and flies on the first dimension due to their expected dissimilarities, its second dimension should indicate whether differently-fed flies regroup into a single cluster (F, left) or whether they cluster away from each other and close to their fruit on this axis (F, right). Regarding expectations (**C, E, G**), all left and right panels are consistent with the '*metabolic generalism*' and '*multi-host metabolic specialism*' hypotheses, respectively.

The online version of this article includes the following figure supplement(s) for figure 1:

**Figure supplement 1.** Principal component analysis (PCA) of fruits' metabolomes.

**Figure supplement 2.** Upset plot of the number of shared ions between all fruits (ten largest intersections).

**Figure supplement 3.** Comparison between the taxonomy of investigated plant species and the biochemical proximity of their fruits.

**Figure supplement 4.** Relation between ion quantity in fruit vs. in flies that developed on the same fruit.

---

variance explained), and cranberry and strawberry by the third dimension of the PCA (25% variance explained). Furthermore, while most of the ions were shared among fruits, qualitative data alone enabled perfect discrimination between fruit, with several hundred ions diagnostic of each investigated fruit (*Figure 1—figure supplement 2*). The distinctive biochemical signatures of the four investigated fruits thus provide ideal ground to decipher the effect of their use on their generalist consumers.

Furthermore, we assessed the proximity of chemical compositions of the investigated fruits using hierarchical clustering of fruit samples based on the Euclidian distance of their scaled ions quantities. A simple visual comparison of topologies between this clustering and the corresponding phylogenetic tree of fruit species indicated that taxonomic classification was not predictive of chemical proximity between fruits (*Figure 1—figure supplement 3*).

We also checked whether fruit-derived ions detected in flies were related to their development on a specific diet (i.e. the metabolization of fruit) or, conversely, may come from direct contamination of the flies by fruits. Since we did not remove the gut material from our processed individuals, nor subjected them to a fasting period, it is possible that remnants of fruit media could lead to contamination. In the case of contamination of our fly samples by fruit diets, we would expect a positive correlation between the ion quantities in fruit samples and those detected in the corresponding fly samples. The absence of direct contamination was confirmed by the absence of correlation between ion quantities in fruits and flies that consumed them (all Kendall's $\tau < 0.2$; *Figure 1—figure supplement 4*). This lack of correlation stemmed both from metabolization or excretion (i.e. decreased amount of ions in the fly relative to the fruit) and accumulation or sequestration (i.e. increased amount of ions in the fly relative to the fruits).

## Host use

Whether individuals of a generalist species process all types of diet in a similar, generic way ('metabolic generalism') or show specific pathways to metabolize unique compounds of different diets ('multi-host metabolic specialism') entails largely non-overlapping expectations for the both qualitative and quantitative biochemical composition of individuals relatively to their diets (*Figure 1*).

The first approach is based on the fact that metabolic generalism is expected to result in particular patterns of overlap between ions when all fruits and flies are considered together (*Figure 1B*). If *D. suzukii* is a metabolic generalist, flies should all share a large part of their ions, fly ions specific to a particular diet should be rare or absent, and/or ions specific to single fruit-fly pair (i.e. a pair of fruit and flies grown over the same fruit) should also be rare (*Figure 1C*, left panel). On the contrary, multi-host metabolic specialism would be detected by any given fruit-fly pair sharing large amounts of ions but not with other pairs, and/or large amounts of ions found only in flies that are specific to each diet (*Figure 1C*, right panel). To test these predictions, we computed intersection sizes between all fruits' and flies' ions from a presence/absence (qualitative) dataset derived from our quantitative *integral* dataset. Comparison of the overall intersections of ions present in the different fruits and the flies

grown over these fruits showed that most ions are shared between large groups of fruits and flies (i.e. with higher degrees of intersection; *Figure 2A*). The largest intersection includes 3691 ions shared by all fruits and all flies (32%). The second largest intersection comprises 1103 ions shared among all flies (and absent in fruits; 10%). The fourth largest intersection contains 412 ions shared among all fruits (and no flies; 4%). High-order degree characterizes all other largest intersections, except for a set of 283 ions uniquely found in blackcurrant fruits. *Figure 2B* focuses on the sets of ions found uniquely in fruits, flies, or fruit-fly pairs (i.e. the intersections of degree 1 and 2). While ions found uniquely in each fruit are not uncommon (>100 ions for all fruits), ions found uniquely in flies raised on a particular diet are rare (two in cherry-fed flies, seven in cranberry-fed flies, and none in strawberry-fed and blackcurrant-fed flies). Ions that are distinctive of a particular fruit-fly pair are also seldom, ranging from 69 in the blackcurrant fruit-fly pair to only seven in the strawberry fruit-fly pair.

Similarity and dissimilarity in the use of different host plants could stem from the metabolic fate of common and unique fruit metabolites when ingested by flies. Differently-fed flies metabolizing compounds that are shared between fruits to produce metabolites shared between flies would be indicative of 'metabolic generalism' (*Figure 1D*). On the opposite, differently-fed flies metabolizing compounds private to a given fruit to produce new metabolites (mostly not shared between flies) would be indicative of 'multi-host metabolic specialism.' We thus inspected the intersection degrees of all fruits and the intersection degrees of all flies, together with their relationships (*Figure 3*). First, the intersection of all fruits (left column) is largely formed from ions common to all fruits (48.6% of all ions present in fruits). Ions present in lower degree intersections are found in approximately equal proportions to each other (intersections of degree 3, 2, and 1 represent 20.1%, 14.8%, and 16.5% respectively). Approximately 10% of all ions found in this study are not detected in fruits, but only in flies (yellow stack, left column). Second, the intersection degrees of flies show marked differences from those of the fruits (*Figure 3*, right column). These include a higher proportion of metabolites common to all flies (76.3% of all ions present in flies), smaller proportions of lower degree intersections (intersections of degrees 3, 2, and 1 represent 10.6%, 4%, and 9% of all ions present in flies, respectively), and a large proportion of fruit metabolites that are absent in flies (19.5% of all detected ions). Third, the inspection of the relationships between fruit and flies' metabolomes shows that most common fruit ions are the largest, but not unique, part of the most common fly ions. It is important to note that 92% of ions that were present in flies but not in any fruit (i.e. derived from de novo produced metabolites; yellow stack, left column) are ions common to all flies regardless of their diet (*Figure 3*, yellow flow from 0 to 4). In contrast to the most common fruit ions, ions unique to a single fruit or shared by a few fruits (fruit intersection degrees 1–3) follow roughly the same distribution of 1/3 being shared by all flies, 1/3 being shared by a few flies, and 1/3 being shared by no flies. This includes ions unique to a given fruit, 39.4% of which are found in all flies. Fly ions that are specific to a given diet come from fruit ions with various intersection degrees. In particular, 27.1% are common to all fruits, 21.5% are unique to a single fruit and only 1.1% are produced de novo (nine ions).

Finally, we visually investigated the multi-dimensional clustering of fruits' and flies' metabolomes using a PCA using the *integral* dataset (*Figure 1F*). If diets are the main influence on flies' metabolomes, flies should not cluster together and rather localize close to their corresponding fruit, especially on the second dimension (*Figure 1G*, right panel). On the contrary, if flies similarly process different diets, flies' metabolomes should cluster together, away from the fruits (*Figure 1G*, left panel). Results of the PCA show that while different fruits are easily discriminated, differently-fed flies all clustered into a single group (*Figure 3—figure supplement 1*).

Altogether, our analyses underscore that the metabolomes of flies fed on different fruits are mostly identical, especially from a qualitative standpoint, and unanimously point to metabolic generalism, over multi-host metabolic specialism, as a mechanism promoting diet breadth.

## Metabolomic discrimination of diets in flies

Given the overall qualitative similarity of flies that developed on different diets, we then investigated whether quantitative hallmarks of diet consumption could be detected in flies. A PCA focusing on the *fly* dataset demonstrated that flies' diets can be readily inferred (*Figure 4*). While the first dimension of the PCA (20% of variance explained) discriminates between biological replicates of our experiment (i.e. generations of flies), the second and third dimensions of the PCA (18.3% and 10.5% of variance explained, respectively) discriminate fly metabolomes according to their diet.

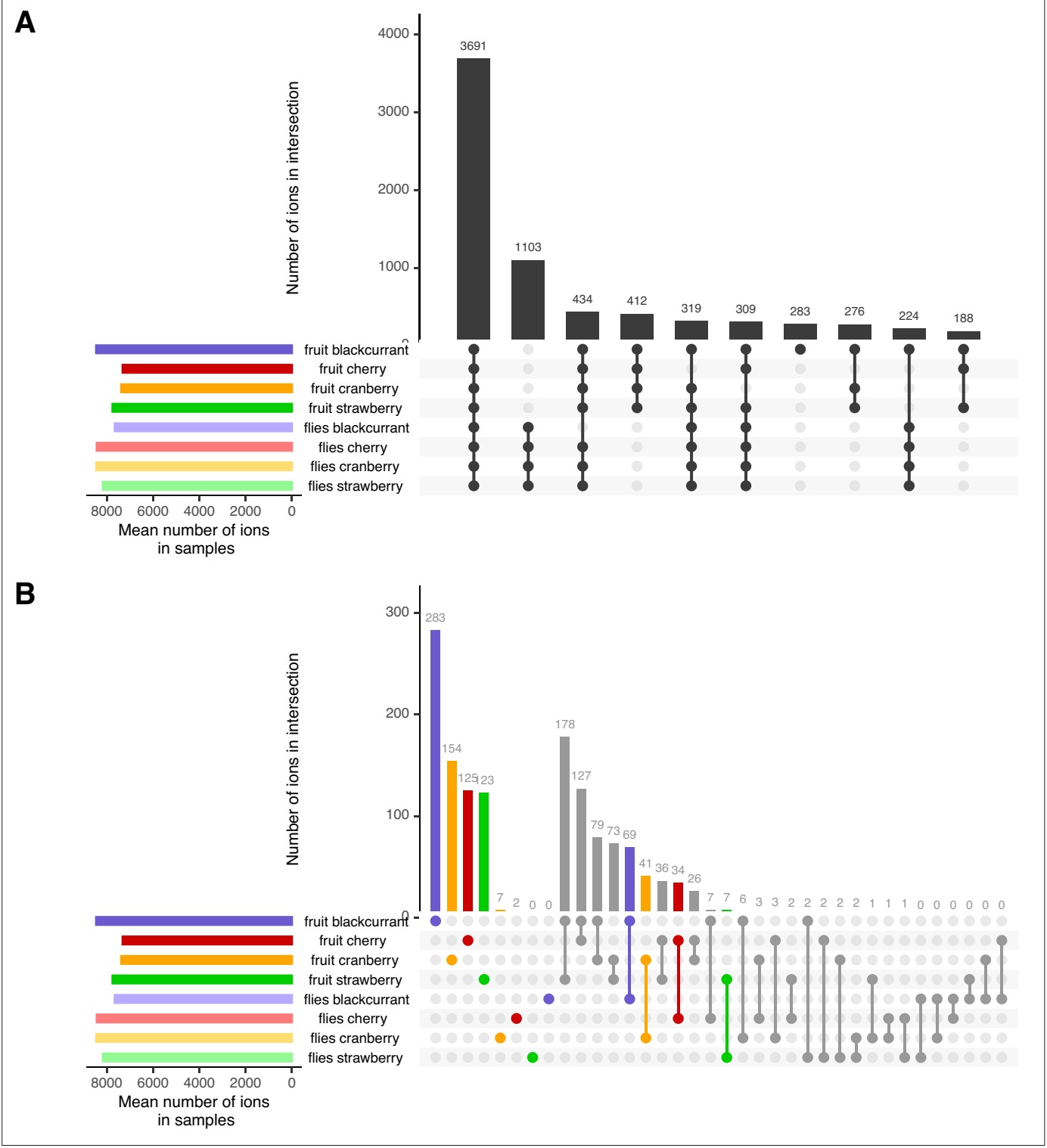

**Figure 2.** Overall trends of shared ions between all fruits and all flies. (**A**) The ten largest intersections between all metabolomes comprise many high-degree intersections, including ions shared between all fruit and all flies' samples (#1), ions, shared between all flies (#2) and ions shared between all fruits (#3). A notable exception to this pattern is the presence of ions found only in the blackcurrant fruits (#5). (**B**) A focus on ions unique to particular samples or sample pairs (i.e. intersections of degree 1 and 2, respectively) illustrate that, while ions unique to fruits are found in high numbers, ions unique to diet in flies are largely absent. Ions specific to a given fruit-flies pair are infrequent. See *Figure 1C* for expectations according to diet generalization hypotheses.

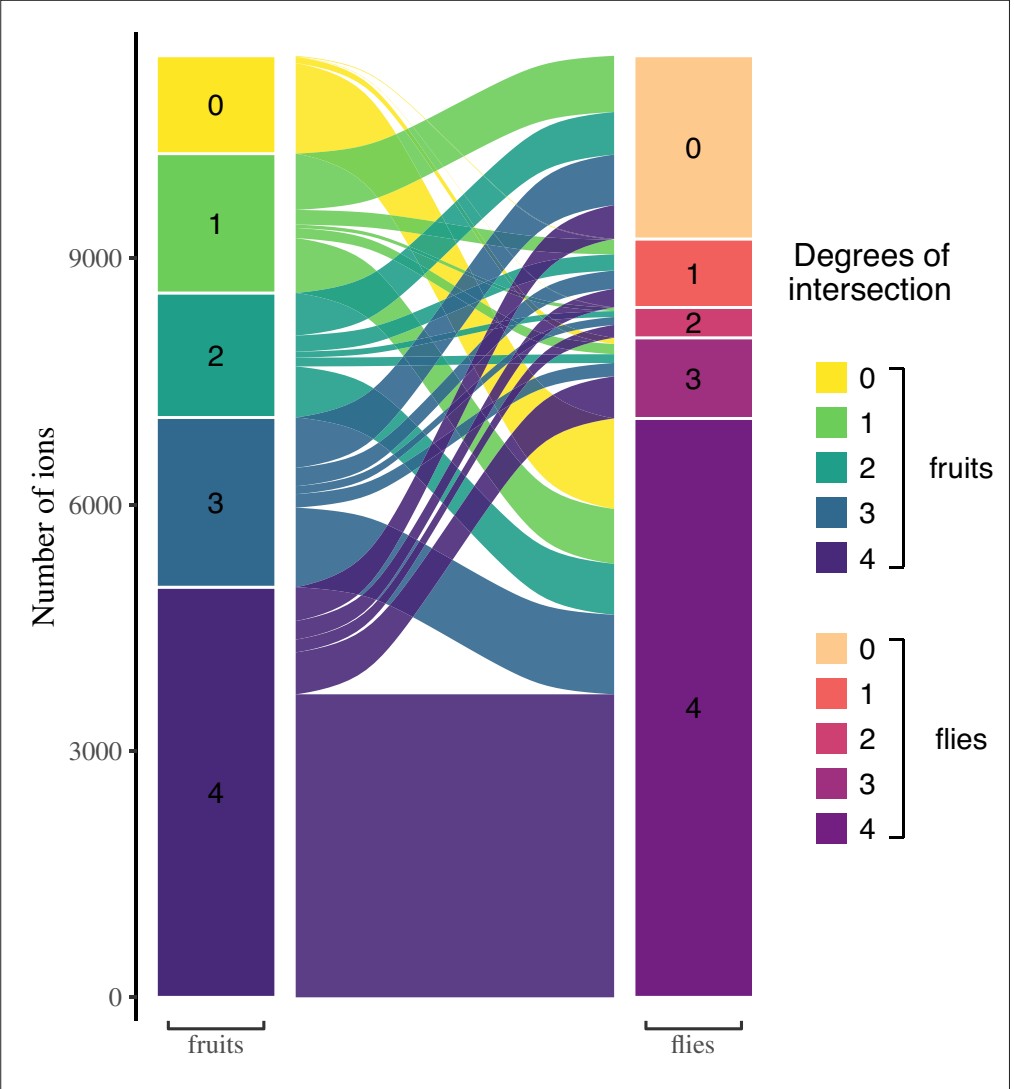

**Figure 3.** Relationships between ions respectively found in fruits and flies, according to their degree of intersection. Ions that are shared within fruits or within flies show a higher degree of intersection. Most ions found in flies are found in all flies, but not necessarily in all fruits. It is also noteworthy that almost all ions produced de novo by flies are found in all flies, regardless of their diet. See *Figure 1E* for expectations according to diet generalization hypotheses.

The online version of this article includes the following figure supplement(s) for figure 3:

**Figure supplement 1.** Fruits and flies' metabolomes cluster separately and do not form pairs.

To further detect which ions were indicative of the development of particular fruits, we then computed diet-specific binomial linear models with Elastic Net regularization. We specifically aimed at producing two lists of diet-specific ions based on their quantitative levels in flies' metabolomes and on two sets of parameters designed (*i*) to optimally classify flies according to their diet ('large' list) and (*ii*) to be as short as possible to allow manual curation of the identified ions ('compact' list, see Materials & methods for details). This statistical approach yielded a 'large' list consisting of 169 fly ions associated with a blackcurrant diet, 103 with a cherry diet, 121 with a cranberry diet, and 40 with a strawberry diet. All four 'large' models enabled a perfect classification of flies according to their diet (ratio of deviance explained >98%). As expected, these ions are part of the top set of ions driving the explained variance in the dimensions of the previous PCA, as shown by their location in the ion space (*Figure 4—figure supplement 1*). The 'compact' set of Elastic Net parameters attained a similar deviance ratio and yielded a list of 62 fly ions associated with a blackcurrant diet, 40 with a cherry diet, 49

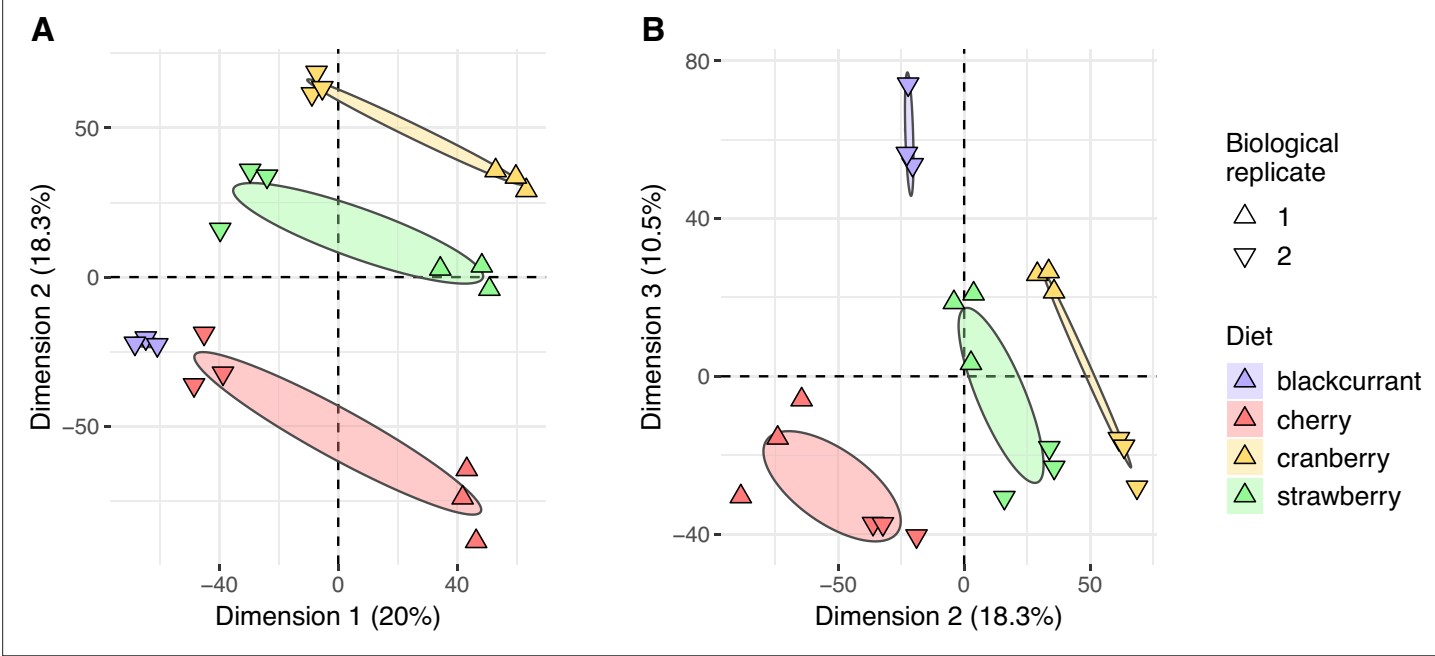

**Figure 4.** Flies' metabolomes differ according to diet. Flies' samples only were included in this principal component analysis (PCA). Differently oriented triangles indicate biological replicates (fly generations). (**A**) Dimensions 1 and 2; (**B**) Dimensions 2 and 3.

The online version of this article includes the following figure supplement(s) for figure 4:

**Figure supplement 1.** Plots of all individual ions of the principal component analysis of fruit metabolomes.

**Figure supplement 2.** Classification of fruit and fly samples following selection of diet-specific ions through GLM with Elastic Net regularization.

**Figure supplement 3.** Quantitative levels of diet-specific fly ions in flies and fruits.

**Figure supplement 4.** Relationships between the quantities of diet-specific fly ions in fruit and flies of the following fruit-fly pairs.

**Figure supplement 5.** Intersection sizes (in fruits and flies) and relationships of fly ions specific to the following diets.

**Figure supplement 6.** Number and degree of intersection of major fruit ions relative to their presence in the consumer flies.

**Figure supplement 7.** Classification performance of qualitative vs quantitative fly datasets to infer diet.

with a cranberry diet, and 15 with a strawberry diet. All fly ions in the 'compact' list were present in the 'large' list. As shown in *Figure 4—figure supplement 2*, the 'large' list of fly ions can perfectly class fly samples according to their diet. It also suggests that most of these fly ions are present in similar quantities in their respective fruit, with few exceptions. Consistent with these results, we verified that ions of the 'large' list for a given flies' diet were significantly more present in flies of the said diet than in flies developed on other diets and also more present in the said fruit than on other fruits (*Figure 4—figure supplement 3*). Depending on the diet, quantities of these distinctive ions found in flies could be correlated to the quantity found in the fruit (e.g. blackcurrant; *Figure 4—figure supplement 4A, B*) or not (e.g. cherry, cranberry, strawberry; *Figure 4—figure supplement 4C, D*).

Diet-associated ions were found to be part of intersections of every degree, both in flies and in fruits (*Figure 4—figure supplement 5*). In flies, large parts of these ions were specific to the flies that consumed the diet, especially for blackcurrant-fed flies (degree 1: 76%, 31%, 33%, and 25% in blackcurrant, cherry, cranberry, and strawberry-fed flies, respectively). However, ions contained in intersections of larger degrees were also observed, including ions found in all flies (degree 4: 16%, 48%, 31%, and 53% in blackcurrant, cherry, cranberry, and strawberry-fed flies, respectively; *Figure 4—figure supplements 5 and 6*). While these diet-distinctive metabolites did not always originate from fruit-specific metabolites and could be detected in other flies and other fruits (*Figure 4—figure supplement 5*), their quantities were lower there than in the focal fruit and flies who consumed it. It is worth noting that few diet-specific ions were produced de novo by flies (fruits intersection of degree zero: 0%, 5%, 5%, and 3% ions in blackcurrant, cherry, cranberry, and strawberry-fed flies, respectively), consistent with the metabolic generalism hypothesis.

Altogether, we found that quantitative levels of ions are critical to correctly classify fly metabolomes according to diet. Consistent with this finding, we observed that qualitative and quantitative datasets did not have the same resolving power when it came to distinguishing flies based on their diet (*Figure 4—figure supplement 7*). While randomly picking ~300 ions would lead to a 75% classification accuracy for flies' diet when using their quantification levels, a similar performance was attained with a sampling of ~1000 ions when using only their presence. Regardless, flies' diet could be readily inferred based on a relatively small subset of ions.

## Identification and fate of metabolites associated with fruit diets

To gain a better understanding of the distinctive features of flies and their diet, we tried to identify diet-specific fly ions of the 'compact' list together with the 50 major ions (i.e. those having the largest peak area) for each investigated fruit. Manual curation and detailed mass spectra analysis allowed us to identify 71 metabolites, 26 of which were confirmed with commercial standards. Identified metabolites belonged to major plant metabolite families, including anthocyanins, flavonoids, organic and hydroxycinnamic acids, and terpenes (*Supplementary file 1*). A two-dimensional clustering analysis of metabolites associated with diet illustrates that even a limited set of these compounds was sufficient to infer flies' diet (*Figure 5*). A survey of the published literature showed that many of the metabolites identified here in specific fruits (and flies that were grown on them) had already been detected and identified in the same fruits (*Figure 5*, *Supplementary file 1*), thereby supporting their correct identification.

Depending on fruits and specific metabolites, relative amounts of fruit-derived metabolites detected in flies did not necessarily mirror their relative abundance in fruits (*Figure 5—figure supplement 1*). Indeed, some major fruit metabolites, such as the anthocyanins delphinidin in blackcurrant, cyanidin in cherry, or pelargonidin in strawberry, were detected in flies, albeit with a relatively low relative abundance (fly/fruit ratio between 0.01 and 0.001). Conversely, other less abundant fruit metabolites, such as terpenes, were relatively abundant in flies (fly/fruit ratio >0.5). Despite the relatively low number of identified metabolites, our quantification results demonstrate that only some fruit metabolites are metabolized (or excreted), while others tend to accumulate in flies.

As expected from previous studies, organic acids tended to be shared by all fruits, but with a particular balance depending on fruits: malic, quinic, and citric acids were always present in fruits (and flies) but with elevated levels in cherry, cranberry, and blackcurrant fruits (and flies that consumed them), respectively. Another chemical family ubiquitous in plants are flavonoids, with many potential physiological effects for consumer organisms, such as neurobehavioral modulation of activity (*Bugel and Tanguay, 2018*), feeding, and oviposition (*Simmonds, 2001*; *Treutter, 2006*). All fruits harbored major flavonoids such as quercetin, kaempferol, or myricetin, with specific substituent groups: quercetin arabinoside and rhamnoside in cranberry, quercetin rutinoside in cherry, quercetin glucoside in blackcurrant and quercetin glucuronide in strawberry. Those substituents have been shown to modulate the bioactivity of flavonoids (*Bugel and Tanguay, 2018*) and might mediate differential developmental responses to fruit diet in *D. suzukii* (*Olazcuaga et al., 2019*). While these substituents can undergo profound modifications by the gut microbiome (e.g. hydrolysis), we found no large-scale trace of microbiota influence in the present study, with flavonoids being essentially carried 'as is' in flies' metabolomes. As an interesting subclass of flavonoids, major anthocyanins characteristic of the investigated fruits (and of their color) were readily found in flies that consumed these fruits. Strawberry-fed flies displayed elevated levels of pelargonidin glucoside, the main anthocyanin of strawberry fruits (*da Silva et al., 2007*). Similarly, cranberry- and cherry-fed flies presented high levels of cyanidin arabinoside, and cyanidin glucoside and rutinoside, the main anthocyanins found in cranberry and cherry, respectively (*Acero et al., 2019*; *Seeram et al., 2004*). Finally, fruits and flies that consumed them also displayed specific chemical signatures associated with terpenoids, considered as the largest class of plant secondary metabolites (*Dudareva et al., 2004*). Our results show that blackcurrant-fed flies can easily be discriminated by their levels in some linalool derivatives. While this volatile, floral-scented monoterpene is found in many plant families (*Knudsen et al., 2006*; *Pichersky and Lewinsohn, 2011*), specific linalool derivatives, e.g., carboxylinalool glycoside and linalooloxide glycoside, were predictive of the flies' blackcurrant diet. Similarly, flies that developed on strawberry can be differentiated based on their elevated content in nerolidol, a sesquiterpene found in the floral scent of many plant families (*Knudsen et al., 2006*) and in fruits including strawberry (*Aharoni et al., 2004*).

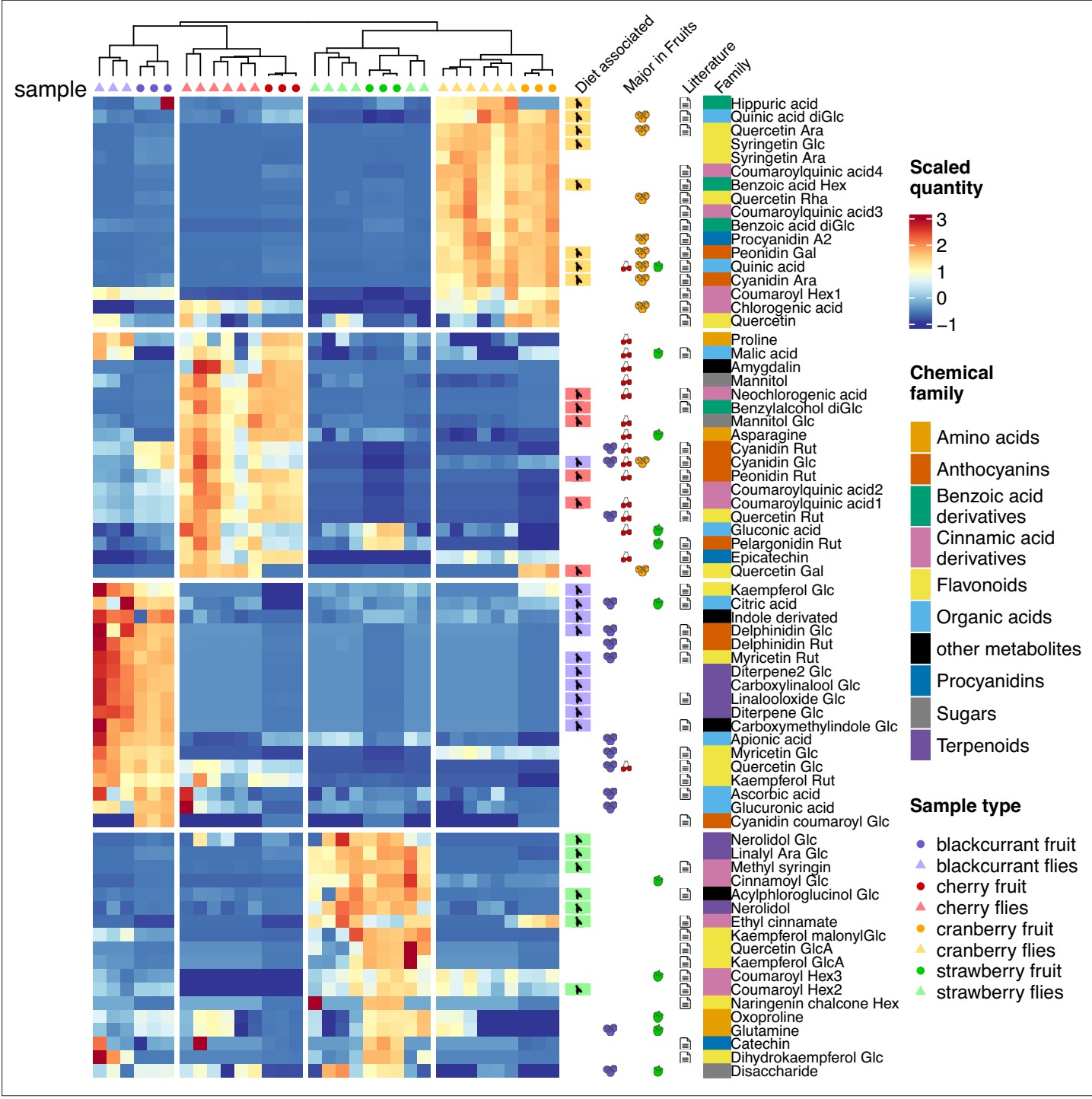

**Figure 5.** Heatmap of the relative quantity of identified metabolites in each fruit and fly sample. Heatmap scale colors indicate the relative quantity of each metabolite (row). Additional columns represent diet specificity in flies, major presence (top 50) in fruits, presence in literature in relation to one or several of the investigated fruits, and chemical family of each identified metabolite (see *Supplementary file 1* for detailed bibliographic information).

The online version of this article includes the following figure supplement(s) for figure 5:

**Figure supplement 1.** Quantities of identified metabolites in the following fruit-fly pairs.

While most of these signatures involve a passive accumulation of plant secondary metabolites, we also uncovered traces of the metabolization of fruit compounds. Indeed, we found elevated levels of hippuric acid in all flies, and particularly in cranberry-fed flies. Hippuric acid originates from the metabolization of various phenolic compounds (such as flavonoids and chlorogenic acids) by the gut microbiome and is a biomarker of fruit (and especially berries) consumption in various species including humans (*De Simone et al., 2021*; *Ulaszewska et al., 2020*). This result highlights that the metabolization process of major fruit compounds by *D. suzukii* may be relatively universal, and lack specificity, as shown by the accumulation of monoterpenes in blackberry-fed flies. However, it is noteworthy that some compounds belonging to other families of toxic plant metabolites do not seem to affect *D. suzukii*. Both cherry puree and cherry-fed flies were found to contain amygdalin, which belongs to the cyanogenic glucoside family. The toxicity of cyanogenic glucosides has been widely documented in the context of plant-insect interactions (*Zagrobelny et al., 2004*). However, cherry has been identified as an excellent host for *D. suzukii*, in both oviposition preference and larval performance experiments (*Olazcuaga et al., 2019*).

## Discussion

The direct comparison of metabolomes of fruits and of individuals that were grown on them sheds some light on basic issues pertaining to diet generalism (*Forister et al., 2012*). Indeed, we found that consumption by generalist individuals of dissimilar diets resulted in a diet-unspecific qualitative response to host biochemistry, i.e., 'metabolic generalism.' This tolerant reaction to diet also leads to the passive accumulation of diet-specific metabolites in individuals. As a result, while individuals were mostly similar across diets, detecting their particular diet was straightforward.

The biochemical composition of stone fruits, berries, and pseudocarps has been explored in numerous studies, highlighting the major contributions of sugars, organic acids, and secondary metabolites to the chemistry of fruits. Our untargeted overview of fruit composition enabled us to confirm that these fruits differ widely in their biochemical composition, and most importantly, that phylogenetic differences correspond to different biochemical landscapes for frugivorous species in our case. It is important to note, however, that the phylogenetic classification of the studied fruit species does not represent their proximity in terms of chemical composition. This strengthens the view that the taxonomic scale may not be the best scale to evaluate diet breadth and the adaptive challenges posed by generalism. For herbivores, it is also important to consider which plant organs are consumed (roots, leaves, fruits), as their respective composition may offer different dietary challenges and different proximities between plant species.

The use of diverse hosts had a variety of biochemical consequences on individuals of a generalist species, all consistent with the 'metabolic generalism' hypothesis. Indeed, diet-specific metabolites were rare, most fruit metabolites were incorporated in all individuals regardless of diet and overall metabolite pattern indicated a proximity across all insect samples. On the one hand, individuals showed a remarkably fixed qualitative chemical response to diet, coherent with a canalized response. This pattern mirrored the metabolic homeostasis found in some generalist insects reared on various protein:carbohydrates (P:C) ratios (*Silva-Soares et al., 2017*; *Watanabe et al., 2019*; *Young et al., 2018*). On the other hand, different diets had a significant impact on the quantity of a multitude of ions, resulting in the straightforward identification of insects' diets based on their metabolome. Our study thus highlights that the physiological response to various diets, while generic in structure, was plastic when relative metabolite amounts were considered. This biochemical response could be expected from studies underlining the transcriptional plasticity in generalist species from all major insect clades (*Birnbaum and Abbot, 2020*; *Celorio-Mancera et al., 2016*; *Celorio-Mancera et al., 2012*; *Celorio-Mancera et al., 2013*; *Christodoulides et al., 2017*; *Ragland et al., 2015*; *Simon et al., 2015*).

The simultaneous observation of a single general qualitative response and diverse quantitative responses to diets is coherent with the metabolic generalism hypothesis and may help reconcile paradoxical notions about the consequences of generalism at the organismal level. On the one hand, individuals from generalist species do not always differ in response to various diets (e.g. *Leptinotarsa decemlineata*, *Forister et al., 2007*; or *Lymantria dispar*, *Barbosa, 1978*; *Lazarevic et al., 1998*). It is a defining characteristic of generalist species that they can tolerate (i.e. display relatively constant developmental responses under) a wider range of dietary circumstances

when compared to specialist species. It makes sense, therefore, that the structural backbone of the metabolome of generalist individuals remains largely unaffected by diet, as uncovered here. On the other hand, the diverse quantitative responses of individuals' metabolomes to diet highlight the passive metabolic response of generalist individuals to the idiosyncrasies of their dietary environment. In particular, we observed that distinctive fruit secondary metabolites, relating to their specific flavor (flavonoids), color (anthocyanins), or scent (terpenoids), were accumulated in corresponding individuals, and that very few de novo diet-specific individuals' metabolites were produced. This may be a common feature in generalist frugivores as they have to face the exceptional diversity of secondary metabolites found in fruits (*Kessler and Kalske, 2018*; *Wetzel et al., 2020*; *Whitehead et al., 2022*).

Our study thus suggests that diet generalization could stem from reduced attention to hosts' biochemical peculiarities and a neutral, passive stance towards potential hosts. While most of the ecological and evolutionary studies of host-plant interactions have focused on intricate plant-insect relations relying on strong selective forces, a passive and generic metabolic response could help species extend their diet breadth, at least transiently on evolutionary timescales. In this case, diet generalism would originate from a general tolerance rather than from the accumulation of independent genetic features associated with the exploitation of different resources (*Forister et al., 2012*; but see *Calla et al., 2017*; *Rane et al., 2019*), and the resulting plastic quantitative response to the diet may not necessarily be adaptive.

Immediate costs of weakened defenses against specific plant compounds can be observed in generalist species (*Ali and Agrawal, 2012*). Here, the passive accumulation of terpenoids in blackcurrant-fed flies indicates that *D. suzukii* may not have evolved efficient monoterpene metabolization pathways. In addition to their attractant or deterrent properties (*Binder and Robbins, 1997*), some terpenes have been shown to exhibit toxic properties and to be subjected to detoxification pathways in insects (*Blomquist et al., 2021*; *Scalerandi et al., 2018*). Indeed, in *D. suzukii*, a blackcurrant diet causes a relatively high larval mortality and leads to the extinction of the population in only a couple of generations (*Olazcuaga et al., 2019*; *Olazcuaga et al., 2021*). However, other fruits harboring potentially harmful compounds, such as amygdalin in cherry, can be excellent hosts for *D. suzukii*, including elevated oviposition preference and larval performance (*Olazcuaga et al., 2019*). Therefore, metabolic generalism may not exclude some adaptations to cope with specific classes of toxic compounds, potentially reflecting the ancestral niche of generalist herbivores.

Metabolic generalism could also procure both immediate and longer-term benefits. For instance, the accumulation of flavonoids increases the fitness of some phytophagous species (e.g. *Polyommatus icarus*; *Simmonds, 2003*). While chemical sequestration is an active process in specialist species, passive mechanisms may be common in generalist species (*Petschenka and Agrawal, 2016*). Here, the probably passive 'sequestration' of flavonoids by generalist individuals might be useful against predators or parasitoids. On larger ecological scales, metabolic generalism could also entail benefits, e.g., by enabling a geographical expansion triggering enemy-release or by reducing intra-guild competition. When successive generations of herbivores must exploit different hosts seasonally (e.g. in frugivores), an immediate reward of metabolic generalism could stem from resource allocation alternatives, e.g., a homeostatic immune response.

The passive accumulation of plant metabolites, such as anthocyanins, may also illustrate a greater opportunity for diet generalism due to plant domestication, especially for frugivorous species (*Whitehead et al., 2017*). Indeed, anthocyanins are known to trade-off with plant defense systems, and the domestication of fruit crops, by acting on anthocyanins accumulation, may have lowered plant defenses and thereby allowed generalist species to exploit niches that were ancestrally too challenging. For instance, it has been demonstrated that the domestication of cranberry cultivars, by selecting upon anthocyanin-related traits such as ripening and fruit color intensity, resulted in lowered plant defenses responses related to the jasmonic acid pathway (*Rodriguez-Saona et al., 2011*). In strawberries, it is known that biochemical insect resistance traits correlate negatively with important agronomical traits such as yield (*Hancock et al., 2008*). On the contrary, blackcurrant domestication mainly focused on plant pathogens and frost resistance for much of its history (*Brennan, 2008*), and this could explain the high levels of detrimental terpenoids in this species. Overall, biochemical changes induced by plant domestication are expected to lead to (*i*) increased diet generalism and (*ii*) increased levels of non-lethal plant secondary metabolites in herbivores, especially in frugivores.

Finally, by showing that fruit-specific metabolites were readily found in metabolomes, our study suggests that metabolomic data might help track the diet of phytophagous insects in the wild (*Alberdi et al., 2019*; *Nielsen et al., 2018*). This could complement high-throughput DNA sequencing-based diet reconstruction, which relies on the presence of foreign DNA material. Prerequisites for achieving this goal include an increased ability to correctly identify plant secondary metabolites and laboratory experiments to study the effect of various diets on the herbivores' metabolomes during development.

In the future, the investigation of different spatial and temporal scales in resource use could help to shed light on the equilibrium between diet generalism and specialization (*Devictor et al., 2010*), and on the origins and limits of diet generalism (*Bolnick et al., 2003*; *Roughgarden, 1972*; *Sexton et al., 2017*). To date, the interaction of phytophagous insect species such as leaf-chewing and mining insects with their host plants has drawn much of the research attention. Compared to more general plant tissues, fruits are specialized organs interacting with multiple species, with detrimental or beneficial impacts to plant fitness, and this may have shaped a greater complexity of the chemical landscape in fruits (*Cipollini and Levey, 1997*). If the response to insect pests tends to decrease the attractiveness or palatability of fruits for more desirable seed dispersers, such as vertebrate frugivores, anti-insect defenses may perturb the reproductive potential of host plants (*Andersen, 1988*; but see *Wilson et al., 2012*). Understanding the ecological constraints exerted on both the chemistry of fruits and the physiology of their consumers, impacting the evolution of niche breadth itself, will require a detailed knowledge of molecular- to community-level interactions across multiple trophic levels.

## Conclusion

This study is a first step toward a finer understanding of the mechanisms supporting the evolution of diet breadth using a joint metabolomic approach of diets and consumers, here underlining the importance of a neutral metabolic generalism. We anticipate more studies will jointly consider plants' and herbivores' metabolisms. This will enable us to gain a deeper understanding of the biochemical diversity of specific plant organs such as fruits and leaves, as well as the physiological origin and maintenance of diet generalism and specialization.

# Materials and methods
## Fruit samples and media preparation

We investigated the influence of four major and economically important host plants of *D. suzukii*: cherry (*Prunus avium*, Rosaceae), cranberry (*Vaccinium macrocarpon*, Ericaceae), strawberry (*Fragaria x ananassa*, Rosaceae), and blackcurrant (*Ribes nigrum*, Grossulariaceae). To focus on their biochemical properties, we used artificial media based on industrial purees of these four fruits, instead of whole fruits, allowing us to avoid the effect of fruit skin, color, size, and shape. Frozen fruit purees were purchased from commercial companies (cherry: Huline S.A.S, Béziers, France; cranberry: Sicodis, Saint Laurent d'Agny, France; strawberry and blackberry: La Fruitière du Val Evel, Evellys, France). Fruit purees were produced only from organic fruits and without added sugars, coloring, or preservatives. We kept the purees frozen until use in order to preserve their original flavors and colors. The use of industrial fruit purees enabled us to attain a homogenous fruit media composition regardless of the ripening periods of each fruit. In other words, this choice allows to study simultaneously all fruits in their optimal states, whatever the timing of the experiment.

To allow the development of *D. suzukii*, fruit media consisted of a mix of 60% (v/v) fruit puree and of 40% (v/v) of a simplified, no-sugar, protein-low version of the 'German Food' medium, an artificial medium classically used in *Drosophila* laboratories. The original recipe of this artificial medium is mainly composed of dry yeast, minerals, sugars, and agar (https://bdsc.indiana.edu/information/recipes/germanfood.html, *Backhaus et al., 1984*). The version of artificial media added to fruit puree is identical to 'German Food', but without any sugar and with 75% of the original protein content (see Table S2 in *Olazcuaga et al., 2019*, for the full recipe and the reference of the products used). These modifications followed two goals: (i) to limit sugar content to the natural portion brought by fruits themselves, and (ii) to keep the protein level low while allowing complete larval development. Fruit media may provide a faithful sensorial representation of fruits in the field. Indeed, *Olazcuaga et al., 2021* have used these same fruit media and have found that oviposition preference and larval performance (traits relevant in host use of a generalist species) were higher in fruit medium corresponding

to the (real) fruit from which the female originated than on fruit media corresponding to other fruits. In addition, even if the choice of fruit media decreased the overall biological relevance of our study, it enabled us to bypass the complex interactions between micro-organisms, fruits, and insects that account for a large part of the protein intake in our model species. The choice of fruit media permitted to allow insect development while controlling protein input and homogenizing fruit quantity and quality. This choice thus constitutes the preferable balance between controlled experimental procedures and ecological relevance. We additionally produced an original 'German Food' medium to control for the putative effect of the artificial (i.e. non-fruit) component of our fruit media on flies' metabolomes (see 'Diet experiment' below).

## Flies

Our *Drosophila suzukii* stock population was initiated by sampling ~1000 flies (more than 500 females and 460 males) in the region of Montpellier (France) in September and October 2016. These flies were collected using baited cup traps (*Lee et al., 2011*). At this time of year and in this area, none of the four fruits studied in this study were available. Based on *Poyet et al., 2015*, flies could have potentially emerged from more than 18 wild host plants, which could be available in the sampling area in October (Table S1 in *Olazcuaga et al., 2021*). This population, therefore, displays a diversity of individuals, none of which was specialized to our four fruits of interest. Before starting the experiment, this population was maintained at a size of ~2000 individuals for a dozen generations in 10 ml flasks neutral medium (consisting of sugar, dry yeast, minerals, and antifungal solution; *Backhaus et al., 1984*), at 21 ± 2°C, 65% relative humidity, and a 16:8 (L:D) hr light cycle in an air-conditioned chamber. Generations were non-overlapping and consisted of 24 hr of oviposition followed by 15 days of development and 6 days of sexual maturation and mating. Each generation all individuals were mixed and then distributed again into 100 flasks of 20 individuals (for more details on the maintenance conditions of this population, see *Olazcuaga et al., 2019*).

## Diet experiment

To study the consequences of a fruit diet on *D. suzukii*, we formed four populations of adults developed on a single fruit medium (cherry, cranberry, strawberry, and blackcurrant) from our stock population. To this aim, 20 groups of 20 six-day-old adults from the stock population (i.e. developed in the neutral medium) were left to mate and oviposit on a single fruit medium for two consecutive periods of 24 hr and then discarded. Ten days after oviposition and immediately after emergence, resulting adults were collected and maintained for seven days on the same fruit medium, with a maximum density of 40 adults per tube. For each fruit medium, these seven-day-old individuals were then flash-frozen at –80 °C in groups of 100 individuals. For each fruit, three groups of 100 flies were stored at –80 °C. This experiment was replicated in the exact same conditions for two consecutive generations of the stock population, in order to obtain biological replicates. During each generation, we also reared a fifth 'control' population that developed exclusively on the artificial component of our final media (i.e. 'German Food' without any fruit puree). We collected and processed this population similarly and simultaneously to the populations that developed on fruits (including two independent generations). This population enabled us to control for artificial medium-derived metabolites in our populations developed on artificial fruit media (see 'Statistical analyses' below).

## Sample preparation

Frozen flies were freeze-dried for 24 hr. Each sample, composed of 30 flies, was weighed. Three samples were collected for each fruit diet and biological replicate (i.e. six samples for each diet), except for blackcurrant flies where only one biological replicate was available (i.e. three samples for blackcurrant flies). Flies were transferred into 2 mL Eppendorf tubes and were crushed in a bead mill (TissueLyser II, Qiagen, Venlo, Netherlands) to obtain a powder. Metabolites were then extracted with methanol (20 µL/mg dry weight) containing 1 µg/mL of chloramphenicol as an internal standard. For fruit purees, 300 mg (fresh weight) was extracted with pure methanol (5 µL/mg fresh weight) containing 1 µg/mL of chloramphenicol. Extractions were performed in triplicate. Samples were placed in an ultrasonic bath (FB15050, Fisher Scientific, Hampton, USA) for 10 min, followed by centrifugation at 12,000 g for 10 min at 4 °C. Supernatants (about 150 µL) were collected and transferred into vials for UHPLC-MS analyses.

## Non-targeted UHPLC-MS analyses

Extracts were analyzed using a UHPLC system (Dionex Ultimate 3000; Thermo Fisher Scientific) equipped with a diode array detector (DAD). Chromatographic separation was performed on a Nucleodur HTec C18 column (150 × 2 mm, 1.8 µm particle size; Macherey-Nagel) maintained at 30 °C. The mobile phase consisted of acetonitrile/formic acid (0.1%, v/v) (eluant A) and water/formic acid (0.1%, v/v) (eluant B) at a flow rate of 0.25 ml/ min. The gradient elution program was as follows: 0–4 min, 80–70% B; 4–5 min, 70–50% B; 5–6.5 min, 50% B; 6.5–8.5 min 50–0% B; 8.5–10 min, 0% B. The injected volume of the sample was 1 µL. The liquid chromatography system was coupled to an Exactive Orbitrap mass spectrometer (Thermo Fisher Scientific) equipped with an electrospray ionization source operating in positive and negative modes. The instruments were controlled with Xcalibur software (Thermo Fischer). The ion transfer capillary temperature was set at 320 °C and the needle voltage at positive and negative modes at 3.4 kV and 3.0 kV, respectively. Nebulization with nitrogen sheath gas and auxiliary gas was maintained at 40 and 5 arbitrary units, respectively. The spectra were acquired within the mass-to-charge ratio ($m/z$) range of 95–1200 atomic mass units (a.m.u.), using a resolution of 50,000 at $m/z$ 200 a.m.u. The system was calibrated internally using dibutyl phthalate as lock mass, giving a mass accuracy of <1 ppm. LC-MS grade methanol and acetonitrile were purchased from Roth Sochiel (Lauterbourg, France), and water was provided by a Millipore water purification system.

## MS-MS analyses

A Vanquish Flex binary UHPLC system (Thermo Scientific, Waltham, MA) coupled with an Exploris 120 Q-Orbitrap MS system (Thermo Scientific, Waltham, MA) were used for the identification of metabolites by data-dependent MS2 (ddMS2). The chromatographic conditions (column, mobile phases, temperature, flow, and injection volume) were the same as for the UHPLC-HRMS analyses described above. The mass spectrometer was operated with a heated electrospray ionization source in positive and negative ion modes. The key parameters were as follows: spray voltage, +3.5 and −3.5 kV; sheath-gas flow rate, 40 arbitrary units (arb. unit); auxiliary-gas flow rate, 10 arb. unit; sweep-gas flow rate, 1 arb. unit; capillary temperature, 360 °C; and auxiliary-gas-heater temperature, 300 °C. The scan modes were full MS with a resolution of 60,000 fwhm (at m/z 200) and ddMS2 with a resolution of 60,000 fwhm; the normalized collision energy was 30 V; and the scan range was m/z 85–1200. Internal mass calibration was operated using EASY-IC internal calibration source allowing single mass calibration for full mass range. Data acquisition and processing were carried out with Xcalibur 4.5 and Free Style 1.7 (Thermo Scientific, Waltham, MA), respectively.

## Processing of metabolomic data

Raw data from all the samples of flies and fruit purees and from methanol as blank control were deposited on the Metabolights repository (*Haug et al., 2020*) under accession number MTBLS4721 (https://www.ebi.ac.uk/metabolights/MTBLS4721). These raw data were converted to the mzXML format (centroid mode) using MSConvert (ProteoWizard). mzXML data were sorted into fourteen classes as follows: four fruit-related data from four host plants (fruit purees), seven fly-related data from groups of flies with host-specific diet, two data from group control flies, and blank (methanol). They were then processed using the XCMS software package (*Smith et al., 2006*). Ion detection parameters of the xcmsSet function of XCMS were as follows: 'centWave,' ppm = 2, noise = 30,000,, mzdiff = 0.001, prefilter = c(5,15000), snthresh = 6, peak width = c(6,35). Peaks were aligned using the obiwarp function using the following group density settings: bw = 10, mzwid = 0.0025, minimum fraction of samples for group validation: 0.5. The fill peaks function was used to integrate the signal in the region where a peak is missing in a sample in order to create a new peak in this sample. Ion identifiers were generated by the XCMS script as MxxxTyyy, where xxx is the $m/z$ ratio and yyy the retention time in seconds. For each ion identifier in each sample, an area corresponding to the integration of the MxxxTyyy peak was generated. All ions present in blank injections (areas in the blanks equal or sup. as those of samples) were suppressed from the datasets. This allowed the alignment of 7.805 peaks for the dataset in the positive mode and 3.665 peaks for the dataset in the negative mode. After specifying the detection mode for each ion (positive or negative), both datasets were merged. Since merging different ionization modes could lead to spurious duplications, we then verified that positive, negative, and merged datasets all gave similar results and identical conclusions. Finally, fruit-related

and fly-related and control-flies data were separated, resulting in a *fruit* dataset, a *fly* dataset, and a control-flies dataset.

## Statistical analyses

All statistical analyses were performed using R v3.6.1 (*R Development Core Team, 2019*). We conducted all analyses presented below on a *fly* dataset that was controlled for artificial diet influence using the composition of 'control' flies' metabolomes (i.e. flies raised on a pure 'German Food' medium, without any fruit). We controlled our *fly* dataset using the following correction: all samples from flies that developed on fruits were centered on the median of the 'control' flies value for each ion and each generation. Following this centering, all samples whose absolute peak area (quantitative) value was <1000 were removed from the controlled *fly* and *integral* datasets. This particular threshold was selected to be close to the practical quantification threshold of the Thermo Exactive mass spectrometer used in this study. This correction enabled us to remove ions whose presence was due to the artificial component of our fruit media (i.e. ions that were similarly expressed in 'control' and 'fruit' flies). We also computed all the subsequent analyses on a raw (i.e. uncontrolled) *fly* dataset. Briefly, the controlled *fly* dataset and the raw *fly* dataset yielded similar results (see Appendix 1). All datasets and analyses presented here can be explored and visualized using a Shiny app at https://fruitfliesmetabo.shinyapps.io/shiny/.

## Non-targeted metabolomic analysis of fruits

To ensure that the selected fruit purees did have different compositions, we performed a PCA on the scaled ion quantities of the *fruit* dataset using the *FactoMineR* R package (*Lê et al., 2008*). To examine the proximity of the biochemical composition of fruits (and compare it visually to the known phylogeny of the corresponding species), we plotted a dendrogram of the optimal hierarchical clustering of our fruit samples, based on the Euclidian distance matrix of scaled ion quantities of the *fruit* dataset, using the *ggtree* (*Yu et al., 2017*) and *fastcluster* (*Müllner, 2013*) R packages. The phylogenetic relationships between blackcurrant (Grossulariaceae, Saxifragales), cranberry (Ericaceae, Asterids), and cherry and strawberry (Rosaceae, Rosids) were drawn from *Zeng et al., 2017*. We additionally plotted the heatmap of the two-dimensional clustering analysis of centered and scaled metabolite quantities of the *fruit* dataset using the *ComplexHeatmap* R package (*Gu et al., 2016*). Seriations of ions (rows) and fruit samples (columns) were performed using the 'traveling salesperson' and the optimal leaf ordering methods of the *seriation* R package, respectively (*Hahsler et al., 2020*).

## Host use

As a preliminary step, we checked whether the ion content of flies was due to direct contamination by fruit by investigating the overall correlation between the quantities of ions in fruit purees and in flies that developed on them. The presence of a correlation would be indicative of contamination, while the absence of a correlation would reveal an absence of contamination of flies by the purees.

To check whether *D. suzukii* flies consume different fruits in a similar (i.e. 'metabolic generalism') or in a dissimilar way (i.e. 'multi-host metabolic specialism'), we used both the *fruit* and *fly* (quantitative) datasets and a presence/absence (qualitative) dataset derived from them. The presence/absence dataset was constructed as follows. First, a given ion replicate ion was considered present if the corresponding peak area following XCMS quantification was >1000. This threshold was selected in order to allow the quantification of low-abundance compounds, as many plant-derived diet compounds were expected to be present in trace amounts in flies. Second, all ions that were present in at least three replicates were considered present, otherwise, they were considered absent. Three types of analyses used this presence/absence dataset.

First, we computed intersection sizes between all fruits' and flies' ions from the presence/absence dataset considered together and investigated these intersections using upset plots (bar plots designed to represent intersections; *UpSetR* R package, *Conway et al., 2017*). Indeed, in the case of a similar use of different diets (metabolic generalism), flies should all share a large part of their ions, fly ions specific to a particular diet should be rare or absent, and/or ions specific to a single fruit-fly pair (i.e. a pair of fruit and flies grown over the same fruit) should also be rare (*Figure 1C*). Reciprocal results would favor the multi-host metabolic specialism hypothesis. For instance, a dissimilar use of fruits

would be detected by any given fruit-fly pair sharing large amounts of ions but not with other pairs, and/or large amounts of ions found only in flies that are specific to each diet.

Second, similarity and dissimilarity in the use of different host plants could stem from the metabolic fate of common and unique fruit ions when ingested by flies. Differently-fed flies metabolizing ions that are shared between fruits to produce ions shared between flies would be indicative of 'metabolic generalism' (*Figure 1E*). On the opposite, differently-fed flies metabolizing ions private to a given fruit to produce new ions (mostly not shared between flies) would be indicative of 'multi-host metabolic specialism'. We thus separately computed the intersections of ions present in all fruits on one hand and in all flies on the other hand. We then related these two datasets to examine (*i*) whether ions most common to all fruits are also common to all flies, (*ii*) whether ions unique to a given fruit are also unique to flies that developed on the same fruit, and (*iii*) what is the origin of diet-specific fly ions. The results of these computations were plotted using the *ggalluvial* R package (*Cory and Read, 2020*). In this analysis, differently-fed flies metabolizing ions that are shared between fruits to produce ions shared between flies would be indicative of metabolic generalism, while differently-fed flies metabolizing ions private to a given fruit to produce new ions (mostly not shared between flies) would be indicative of multi-host metabolic specialism (*Figure 1E*).

Finally, we visually investigated the multi-dimensional clustering of fruits' and flies' metabolomes using an overall PCA with the (quantitative) *fruit* and *fly* datasets (*Figure 1F*). Metabolic generalism ought to cause all flies' metabolomes to cluster together, regardless of diet, while multi-host metabolic specialism would be detected if fruit-fly pairs of metabolomes cluster together (*Figure 1G*).

## Metabolomic discrimination of flies according to their diets

To investigate the putative discrimination of flies according to their diet, we first conducted a PCA analysis on the scaled ion quantities of the large *fly* dataset, using the *FactoMineR* R package (*Lê et al., 2008*). Following the results of the previous PCA (see Results section), we computed a list of fly ions most associated with each diet using generalized linear models with Elastic Net regularization (*Zou and Hastie, 2017*), as implemented in the *glmnet* R package (*Friedman et al., 2010*). Elastic Net regularization is particularly useful to distinguish between causal features (in our case, ions) between treatments when the number of samples is small in comparison with the number of features (*Kirpich et al., 2018*). The Elastic Net regularization combines (in proportion $\alpha$) LASSO and ridge regression methods to build on their respective strength and detect significant features in highly correlated spaces, such as found in ~omics data (*Acharjee, 2013*; *Gonzales and De Saeger, 2018*; *Waldmann et al., 2013*). For each diet, all fly samples were used and classified as either from the given diet or not. Preliminary exploration of Elastic Net mixing parameter $\alpha$ and regularization parameter $\lambda$ using 10-fold cross-validation with the *caret* R package (*Kuhn, 2021*) indicated that the method provided good sample classification using various combinations of $\alpha$ and $\lambda$. Two sets of parameters were then used following two distinct goals. The 'large' set of parameters ($\alpha$ = 0.5 and $\lambda$ = 0.01, except $\lambda_{blackcurrant}$ = 0.001) was constructed to enable optimal classification of flies according to diets. The 'compact' set of parameters ($\alpha$ = 0.9 and $\lambda$ = 0.001, except $\lambda_{blackcurrant}$ = 0.0001) was modified to enable manual curation of the list of metabolites significantly associated with diet, by reducing its overall length.

We plotted the heatmap of the two-dimensional clustering analysis of scaled quantities of the flies' ions present on the 'large' list for both flies and fruits to simultaneously visualize (*i*) the classification performance of flies according to their diet and (*ii*) the relative quantities of these flies' ions in every fruit, using the *ComplexHeatmap* R package (*Gu et al., 2016*). We tested whether ions of the 'large' list for a given diet were significantly more present in the flies of the same diet than in flies of alternate diets, and in the same fruit than in alternate fruits. To this aim, we used a bootstrap resampling procedure (n=1000 replicates) to estimate median quantities of ions in all flies and fruits and their 95% confidence intervals, as recommended for heavily skewed continuous data containing zero values (*Helsel, 2011*). For each diet, ion quantities in flies were plotted against the ion quantities in their respective fruit (on a log scale), and their correlation (and associated $R^2$) was computed through linear modeling. Additionally, we investigated the intersections of origin (and their degrees) of these diet-specific ions, both in fruits and flies, to gain insight about their specificity.

Finally, to evaluate the respective power of our qualitative and quantitative datasets to correctly classify flies according to their diet, we randomly sampled ions in batches of various sizes (10–10,000 ions; each random sampling repeated 10,000 times). We then computed (*i*) the proportion of flies

correctly classified to their diet and (*ii*) Adjusted Mutual Information (AMI, a comparison index between clustering; *Vinh et al., 2010*) for each sample size for both quantitative and qualitative datasets. AMI was computed using the *aricode* R package (*Chiquet et al., 2020*).

## Identification of metabolites

We attempted to identify two different categories of metabolites. First, we tried to identify metabolites from the compact' list of ions associated with diet in flies (from our Elastic Net analysis). Second, we investigated 50 major ions in each fruit (i.e. with the highest peak area), potentially corresponding to major metabolites in each fruit.

Metabolites were identified based on molecular formula generated from accurate mass measurements, given the fact that mass spectrometry conditions were chosen to obtain fragments by in-source fragmentation, in addition to pseudo-molecular ions. In addition, MS/MS analyses were performed with all samples for fragmentation-based metabolite annotation. Metabolite identifications were proposed based on expertized analysis of the corresponding mass spectra and fragmentation patterns, in comparison with authentic standards when available, with the MassBank (https://massbank.eu/Mass-Bank/), mzCloud (https://www.mzcloud.org/), and Chem-Spider (http://www.chemspider.com/) mass spectral databases as well as with the published literature. Following this process, an identification confidence level was assigned to all identified metabolites, according to *Schymanski et al., 2014*. This has allowed the putative identification of 71 food-associated metabolites detected in flies, 30 of which were confirmed with authentic standards. Finally, for each identified compound in each fruit, a survey of the published literature was performed to check for previous identification of this compound in this fruit. Metabolite identification information as well as related literature references are presented in *Supplementary file 1*. Standards for metabolite identification were purchased from Sigma-Aldrich (Saint-Quentin Fallavier, France) and Extrasynthese (Lyon, France).

Two-dimensional clustering analyses of the scaled quantities of the 71 identified compounds were computed overall fruit and fly samples, using ions selected from the 'compact' list and major fruit ions, and visualized using the *ComplexHeatmap* R package (*Gu et al., 2016*).

## Acknowledgements

JF was supported by recurrent institutional funding from INRAE. JF would like to thank Aude Gilabert, Pierre E Sirius, and Ariane N Vega for their outstanding support. We thank Benoît Facon and Réjane Streiff for their comments on the manuscript. LO acknowledges support from the European Union program FEDER FSE IEJ 2014–2020 (project CPADROL), the INRAE scientific department SPE (AAP-SPE 2016), and the US National Science Foundation (DEB-1930650 to Ruth Hufbauer).

## Additional information

### Funding

| Funder | Grant reference number | Author |
|---|---|---|
| European Regional Development Fund | FEDER FSE IEJ CPADROL | Laure Olazcuaga |
| INRAE scientific department SPE | AAP-SPE 2016 | Laure Olazcuaga |
| National Science Foundation | DEB-1930650 | Laure Olazcuaga |
| INRAE, France's National Research Institute for Agriculture, Food and Environment | | Julien Foucaud |

The funders had no role in study design, data collection and interpretation, or the decision to submit the work for publication.

## Author contributions
Laure Olazcuaga, Conceptualization, Resources, Funding acquisition, Investigation, Methodology, Writing – original draft, Writing – review and editing; Raymonde Baltenweck, Conceptualization, Resources, Data curation, Validation, Investigation, Methodology, Writing – original draft, Writing – review and editing; Nicolas Leménager, Conceptualization, Resources, Investigation, Methodology, Writing – original draft, Writing – review and editing; Alessandra Maia-Grondard, Conceptualization, Resources, Validation, Investigation, Methodology; Patricia Claudel, Conceptualization, Resources, Investigation, Methodology; Philippe Hugueney, Conceptualization, Resources, Supervision, Funding acquisition, Validation, Investigation, Methodology, Writing – original draft, Project administration, Writing – review and editing; Julien Foucaud, Conceptualization, Resources, Software, Formal analysis, Supervision, Funding acquisition, Validation, Investigation, Visualization, Methodology, Writing – original draft, Project administration, Writing – review and editing

## Author ORCIDs
Laure Olazcuaga http://orcid.org/0000-0001-9100-1305
Raymonde Baltenweck http://orcid.org/0000-0002-8228-1517
Julien Foucaud http://orcid.org/0000-0003-2272-3149

## Decision letter and Author response
Decision letter https://doi.org/10.7554/eLife.84370.sa1
Author response https://doi.org/10.7554/eLife.84370.sa2

---

# Additional files

## Supplementary files
• MDAR checklist

• Supplementary file 1. Table of all identified metabolites, their physical properties, presence in fruits and flies (this study), and previous identifications in fruits.

## Data availability
All data & code generated during this study have been deposited in the INRAE dataverse. The shiny application enabling the exploration and analysis of our complete dataset (PCA, GLM/Elastic Net and associated visualizations) is available here.

The following dataset was generated:

| Author(s) | Year | Dataset title | Dataset URL | Database and Identifier |
|---|---|---|---|---|
| Foucaud J, Olazcuaga L, Baltenweck R, Leménager N, Maia-Grondard A, Claudel P, Hugueney P | 2022 | DATA and CODE for "Metabolic consequences of various fruit-based diets in a generalist insect species" | https://doi.org/10.57745/G4D3PG | Dataverse INRAE, 10.57745/G4D3PG |

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

## Appendix 1

### Analysis of the raw *fly* dataset

In the main text, analyses of qualitative and quantitative aspects of the flies' metabolomes have been conducted on a *fly* dataset controlled for the presence of an artificial component (mainly yeast) in our fruit-based media. Since this artificial component was kept low and identical between fruit media, it may be argued that the analysis of the raw (without correction) *fly* dataset could be informative of the robustness of our conclusions. We here undertake the same analyses than the main text, but using the raw *fly* datasets in place of the controlled *fly* dataset. The raw *fly* dataset is the *fly* dataset obtained at the end of the '*Processing of metabolomic data*' section of Materials and methods, before any correction. We show below that a raw *fly* dataset produced similar results when compared to our controlled *fly* datasets, all patterns being consistent and figures being only marginally different. Our conclusions about diet generalism are therefore robust to the presence of artificially-derived metabolites.

Analyses conducted on the raw *fly* dataset are identical to those performed on the controlled *fly* dataset in the main text. Preliminarily, we checked whether ions content of flies was due to direct contamination by fruit by investigating the overall correlation between the quantities of ions in fruit purees and in flies that developed on them. Then, we inferred host use in relation to the 'metabolic generalism' and 'multi-host metabolic specialism' hypotheses in several steps: (i) construction of a presence/absence dataset using the same procedure as in the main text (See Materials and methods, '*Host use*' section), (ii) examination of intersection sizes between all fruits' and flies' ions from the presence/absence dataset, (iii) computation of the relationships between the intersections of ions present in all fruits on one hand and in all flies on the other hand, and (iv) investigation of the multi-dimensional clustering of fruits' and flies' metabolomes using an overall PCA with the (quantitative) *fruit* and raw *fly* datasets. Similarly to the analyses performed on the controlled *fly* dataset, we next investigated the metabolomic discrimination of flies according to their diets, by computing a list of fly ions most associated with each diet using generalized linear models with Elastic Net regularization (using the same parameters). Using the inferred 'large' list, we undertook identical analyses as presented in the main text: we (i) constructed a heatmap to visualize the classification performance of flies according to their diet and the relative quantities of these flies' ions in every fruit, (ii) checked whether ions of the 'large' list for a given diet were significantly more present in the flies of the same diet than in flies of alternate diets, and in the same fruit than in alternate fruits, (iii) investigated the intersections of origin (and their degrees) of these diet-specific ions, both in fruits and flies, using diet-specific alluvial plots, and (iv) evaluated the respective power of our qualitative and quantitative (raw) datasets to correctly classify flies according to their diet using an identical resampling procedure (see Materials and methods section, '*Metabolomic discrimination of flies according to their diets*' for details). Names of all figures both for raw and controlled dataset are provided in *Appendix 1—table 1*.

### Results

Similarly to previous results using the controlled *fly* dataset, absence of direct contamination was confirmed by the absence of correlation between ion quantities in fruits and flies that consumed them using the raw *fly* dataset (all Kendall's $\tau<0.2$; *Appendix 1—figure 1*).

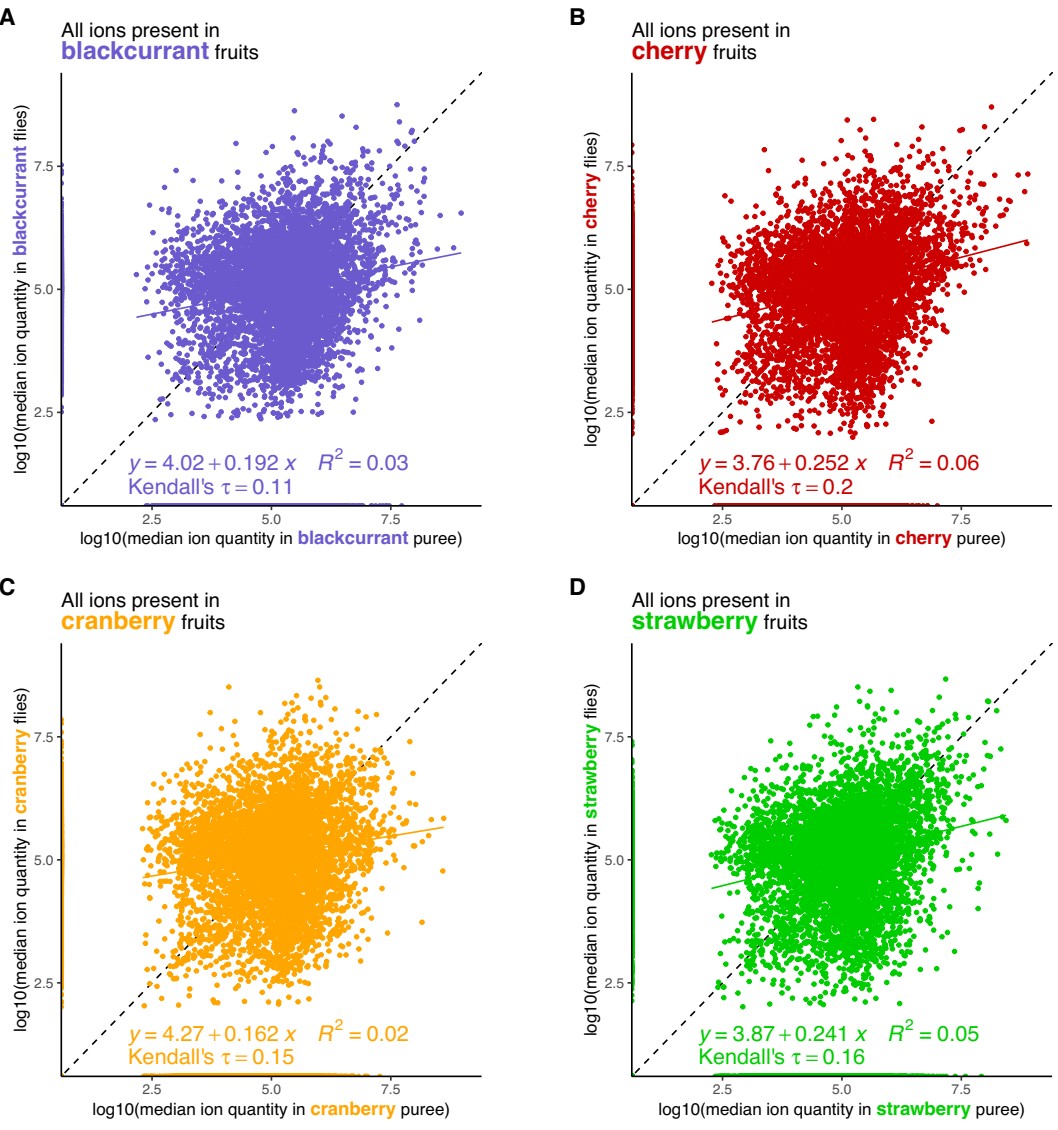

**Appendix 1—figure 1.** Relation between ion quantity in fruit vs in flies that developed on the same fruit. (**A**) Blackcurrant, (**B**) Cherry, (**C**) Cranberry, (**D**) Strawberry. Linear regression coefficients, R-square and Kendall's $\tau$ are indicated for each fruit-fly pair.

## Host use

Whether individuals of a generalist species process all types of diet in a similar, generic way ('metabolic generalism') or show specific pathways to metabolize unique compounds of different diets ('multi-host metabolic specialism') entails largely non-overlapping expectations for both qualitative and quantitative biochemical composition of individuals relatively to their diets (see main text, *Figure 1*). As in main text, we computed intersection sizes between all fruits' and flies' ions from a presence/absence (qualitative) dataset derived from our quantitative integral dataset to test our predictions. Comparison of the overall intersections of ions present in the different fruits and the flies grown over these fruits (using the raw *fly* dataset) showed that most ions are shared between large groups of fruits and flies (i.e. with higher degrees of intersection; *Appendix 1—figure 2A*). The largest intersection includes 3754 ions shared by all fruits and all flies (33%). The second largest intersection comprises 1030 ions shared among all flies (and absent in fruits; 9%). The third largest intersection contains 463 ions shared among all fruits (and no flies; 4%). High-order degree characterizes all other largest intersections, except for a set of 297 ions uniquely found in blackcurrant fruits. *Appendix 1—figure 2B* focuses on the sets of ions found uniquely in fruits, flies or fruit-fly pairs (i.e. the intersections

of degree 1 and 2). While ions found uniquely in each fruit are not uncommon (>100 ions for all fruits), ions found uniquely in flies raised on a particular diet are rare (nine in cherry-fed flies, six in cranberry-fed flies, one in blackcurrant-fed flies, and none in strawberry-fed flies). Ions that are distinctive of a particular fruit-fly pair are also seldom, ranging from 77 in the blackcurrant fruit-fly pair to only 12 in the strawberry fruit-fly pair. The overall pattern of these results is identical to results obtained using the controlled *fly* dataset, with figures being only marginally smaller (e.g. 3691 ions present in the largest intersection when using the controlled *fly* dataset).

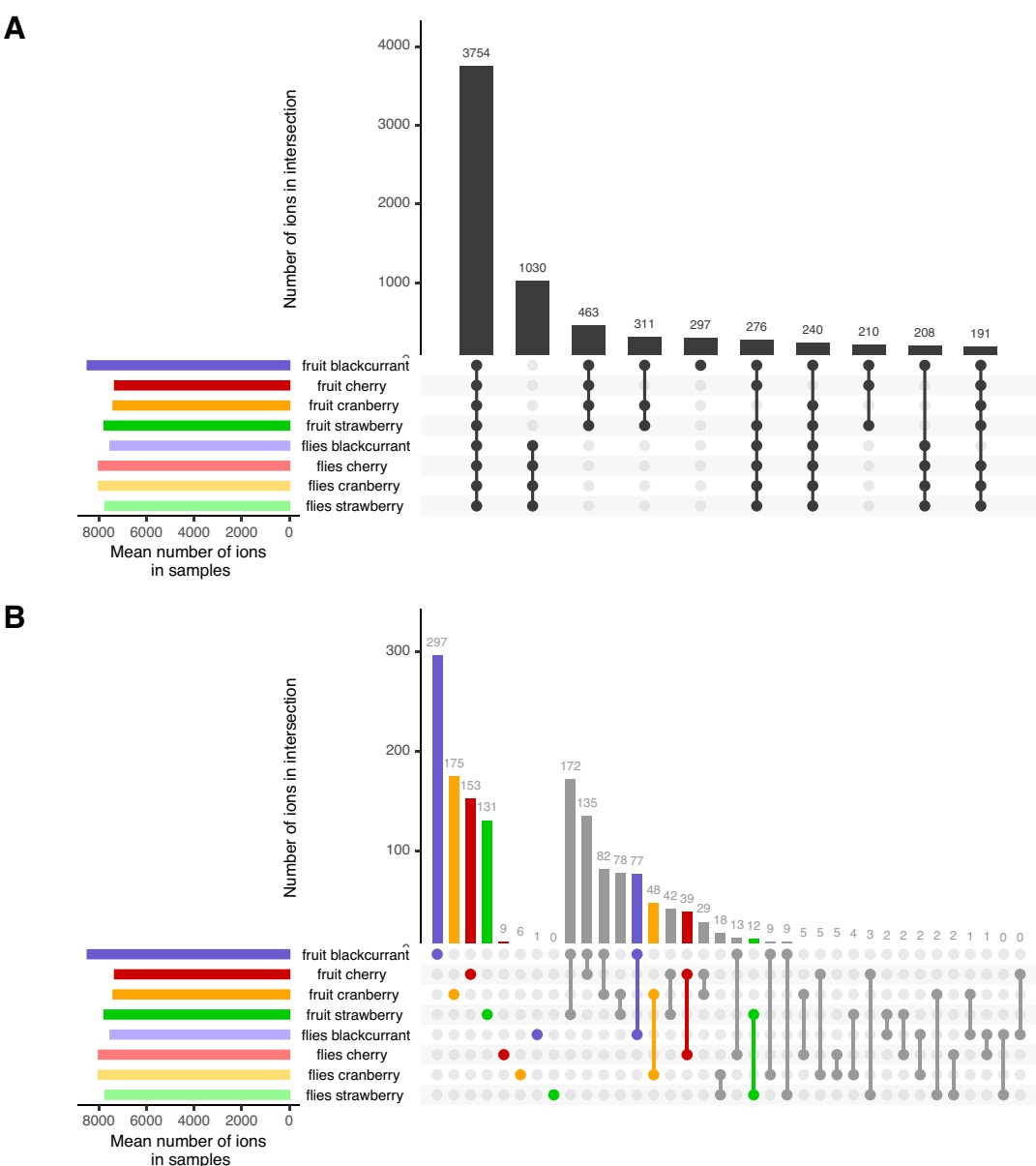

**Appendix 1—figure 2.** Overall trends of shared ions between all fruits and all flies. (**A**) The ten largest intersections between all metabolomes comprise many high degree intersections, including ions shared between all fruit and all flies' samples (#1), ions, shared between all flies (#2) and ions shared between all fruits (#3). A notable exception to this pattern is the presence of ions found only in the blackcurrant fruits (#5). (**B**) A focus on ions unique to particular samples or sample pairs (i.e. intersections of degree 1 and 2, respectively) illustrate that, while ions unique to fruits are found in high numbers, ions unique to diet in flies are largely absent. Ions specific to a given fruit-flies pair are infrequent. See *Figure 1C* for expectations according to diet generalization hypotheses.

We next inspected the intersection degrees of all fruits and the intersection degrees of all flies, together with their relationships (*Appendix 1—figure 3*), to verify predictions made in *Figure 1D* of

the main text. First, the results for intersections of all fruit are identical to the results provided in the main text (i.e. the *fruit* dataset is identical). Namely, the intersection of all fruits (left column) is largely formed from ions common to all fruits (48.6% of all ions present in fruits). Ions present in lower-degree intersections are found in approximately equal proportions to each other (intersections of degrees 3, 2, and 1 represent 20.1%, 14.8%, and 16.5%, respectively). Approximately 10% of all ions found in this study are not detected in fruits, but only in flies (yellow stack, left column). Second, as showed with the controlled *fly* dataset, the intersection degrees of flies (with the raw dataset) show marked differences from those of the fruits (*Appendix 1—figure 3*, right column). These include a higher proportion of metabolites common to all flies (74.7% of all ions present in flies in the raw *fly* dataset vs. 76.3% in the controlled *fly* dataset), smaller proportions of lower-degree intersections (intersections of degrees 3, 2, and 1 represent 7.8%, 6.1%, and 11.3% of all ions present in flies, respectively; vs. 106%, 4% and 9% with the control *fly* dataset), and a large proportion of fruit metabolites that are absent in flies (21.2% of all detected ions with the raw *fly* dataset vs. 19.5% with the controlled *fly* dataset). Third, the inspection of the relationships between fruit and flies' metabolomes shows that most common fruit ions are the largest, but not unique, part of the most common fly ions. It is important to note that 86% of ions that were present in flies but not in any fruit (i.e. derived from de novo produced metabolites; yellow stack, left column) are ions common to all flies regardless of their diet (*Appendix 1—figure 3*, yellow flow from 0 to 4). In contrast to the most common fruit ions, ions unique to a single fruit or shared by a few fruits (fruit intersection degrees 1–3) follow roughly the same distribution of 1/3 being shared by all flies, 1/3 being shared by a few flies, and 1/3 being shared by no flies. This includes ions unique to a given fruit, 35.2% of which are found in all flies. Fly ions that are specific to a given diet come from fruit ions with various intersection degrees. In particular, 25.7% are common to all fruits, 22.6% are unique to a single fruit and only 1.6% are produced de novo (16 ions). These figures are similar to the pattern drawn by the analysis of the controlled *fly* dataset (39.4% of unique fruit ions found in all flies; 27.1% of diet-specific fly ions found in all fruit, 21.5% are unique to a single fruit, and only 1.1% are produced de novo).

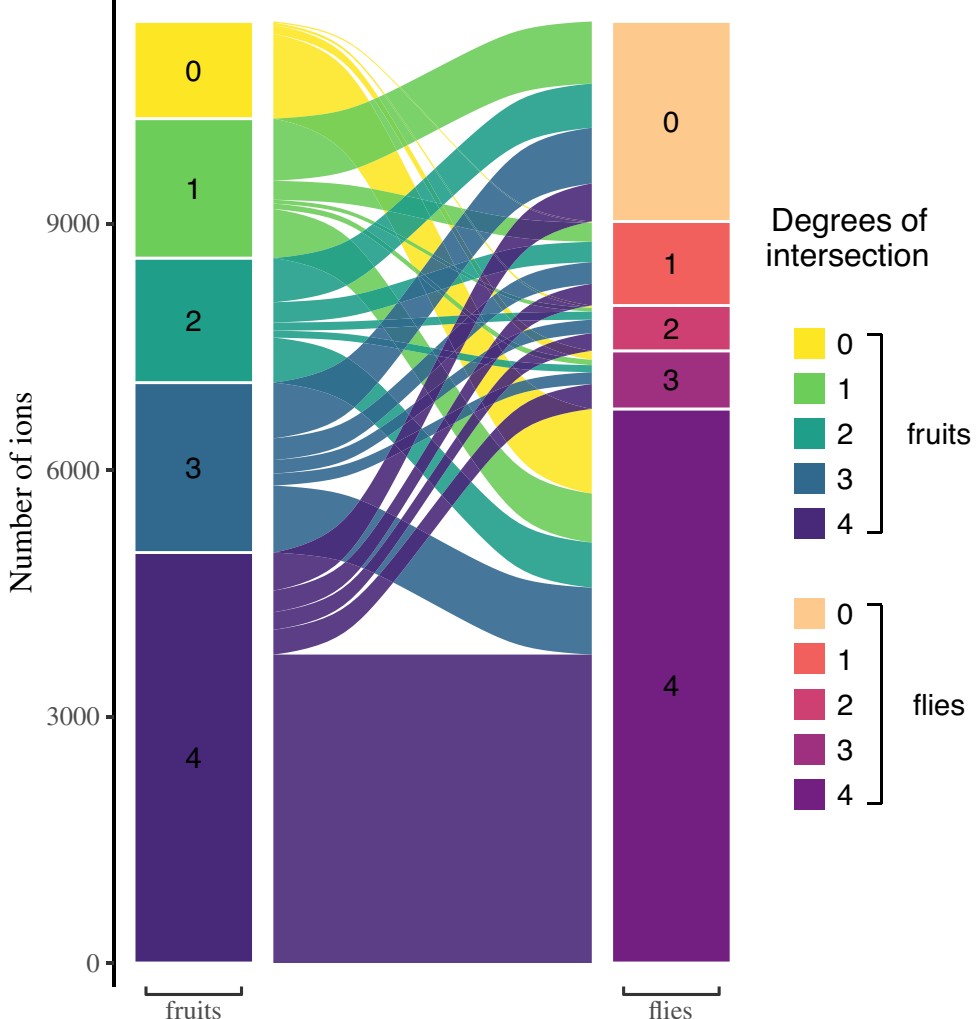

**Appendix 1—figure 3.** Relationships between ions respectively found in fruits and flies, according to their degree of intersection. Ions that are shared within fruits or within flies show a higher degree of intersection. Most ions found in flies are found in all flies, but not necessarily in all fruits. It is also noteworthy that almost all ions produced de novo by flies are found in all flies, regardless of their diet. See *Figure 1E* for expectations according to diet generalization hypotheses.

Finally, we visually investigated the multi-dimensional clustering of fruits' and flies' metabolomes using a PCA using the raw *integral* dataset (joining the *fruit* dataset and the raw *fly* dataset). If diets are the main influence on flies' metabolomes, flies should not cluster together and rather localize close to their corresponding fruit (main text, *Figure 1G*, right panel). On contrary, if flies similarly process different diets, flies' metabolomes should cluster together, away from the fruits (main text, *Figure 1G*, left panel). Results of the PCA show that while different fruits are still easily discriminated, differently-fed flies all clustered into a single group (*Appendix 1—figure 4*).

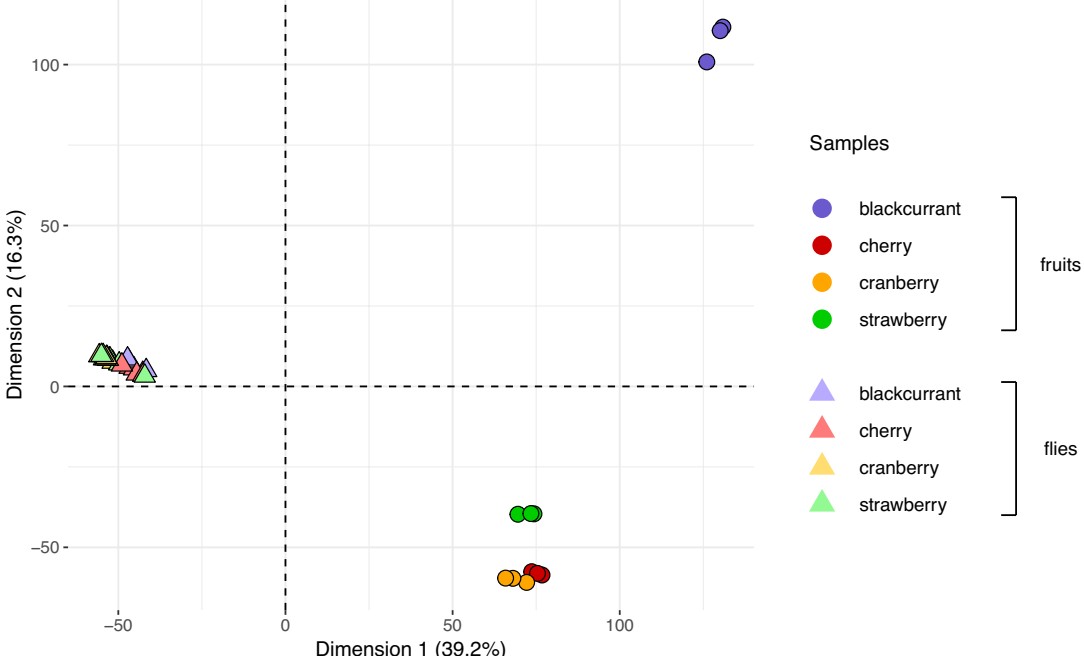

**Appendix 1—figure 4.** Fruits and flies' metabolomes cluster separately and do not form pairs. Fruit and fly samples in this overall principal component analysis (PCA) are represented by dots and triangles, respectively. Colors represent different fruits or diets in the case of flies. All flies' samples cluster together and away from fruits, regardless of their diets. See *Figure 1G* for expectations according to diet generalization hypotheses.

## Metabolomic discrimination of diets in flies

Given the overall qualitative similarity of flies that developed on different diets, we then investigated whether quantitative hallmarks of diet consumption could be detected in flies. A PCA focusing on the raw *fly* dataset demonstrated that flies' diet can be readily inferred (*Appendix 1—figure 5*). While the first dimension of the PCA (21.6% of variance explained) discriminates between biological replicates of our experiment (i.e. generations of flies), the second and third dimensions of the PCA (17.5% and 9.4% of variance explained, respectively) discriminate fly metabolomes according to their diet. These results mirror those obtained using the controlled *fly* dataset, which also showed a first dimension driven by the generation of flies, followed by the second and third dimensions enabling perfect clustering of flies according to diet (main text, *Figure 4*). We notice that the control *fly* dataset enabled to gain slightly more signal from diet (18.3% and 10.5% of variance explained, respectively) and less from generation (20% of variance explained), without altering the conclusions.

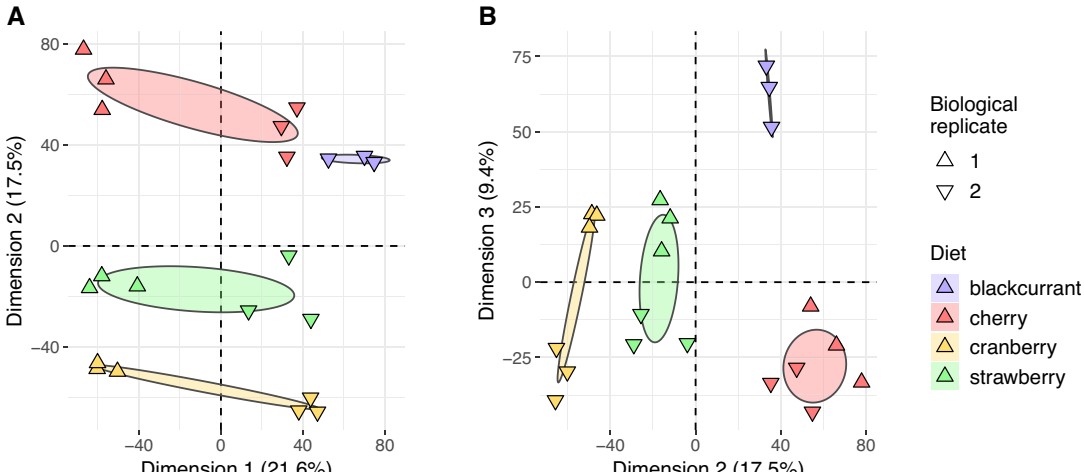

**Appendix 1—figure 5.** Flies' metabolomes differ according to diet. Flies' samples only were included in this principal component analysis (PCA). Differently oriented triangles indicate biological replicates (fly generations). (**A**) Dimensions 1 and 2; (**B**) Dimensions 2 and 3.

The GLM Elastic Net analysis of the raw *fly* dataset resulted in marginally modified lists of diet-specific ions when compared to the controlled *fly* dataset: the 'large' and 'compact' lists for each fruit did not differ by more than 10 ions. The 'large' list resulting from the analysis of the raw *fly* dataset consisted in 162 (vs. 169) fly ions associated with a blackcurrant diet, 98 (vs. 103) with a cherry diet, 111 (vs. 121) with a cranberry diet, and 37 (vs. 40) with a strawberry diet. The 'compact' list of diet-specific ions resulting from the analysis of the raw *fly* dataset consisted in 64 (vs. 62) fly ions associated with a blackcurrant diet, 38 (vs. 40) with a cherry diet, 46 (vs. 49) with a cranberry diet, and 12 (vs. 15) with a strawberry diet. Uncovered diet-specific ions were mostly identical when using raw or controlled *fly* datasets (blackcurrant 'large': 92%; blackcurrant 'compact': 87%; cherry 'large': 86%; cherry 'compact': 83%; cranberry 'large': 87%; cranberry 'compact': 84%; strawberry 'large': 75%; strawberry 'compact': 67%). In spite of these minor differences, diet-specific ions uncovered using the raw *fly* dataset clustered in the same areas of the ions' PCA (*Appendix 1—figure 6*) and enabled perfect classification of flies according to their diet (*Appendix 1—figure 7*). In line with the results using the controlled *fly* dataset, diet-specific ions uncovered here using the raw *fly* dataset were significantly more present in flies of the said diet than in flies developed on other diets and also more present in the said fruit than on other fruits (*Appendix 1—figure 8*). Depending on the diet, quantities of these distinctive ions found in flies could be correlated to the quantity found in the fruit (e.g. blackcurrant, cherry; *Appendix 1—figure 9*) or not (e.g. cranberry, strawberry; *Appendix 1—figure 9*). In flies, analysis of the raw dataset confirmed that both ions contained only in the fruits consumed of in intersections of larger degrees could be observed (*Appendix 1—figures 10 and 11*). Again, classification of flies according to their diet using quantitative data outperformed classification based on qualitative data alone (*Appendix 1—figure 12*).

We wish to underline here that the only minor difference between all analyses of the raw and controlled fly dataset is the quantities of cherry-specific ions in flies being relatively more correlated to their quantities in fruit when using the raw *fly* dataset. This result is anecdotal and does not modify our conclusions regarding diet generalism. 'Compact' lists were always completely included in 'large' lists of diet-specific ions. The result of slightly shorter lists of diet-specific ions when using the raw *fly* dataset is consistent with our previous observation of a marginally better fruit signal in the control dataset (as expected).

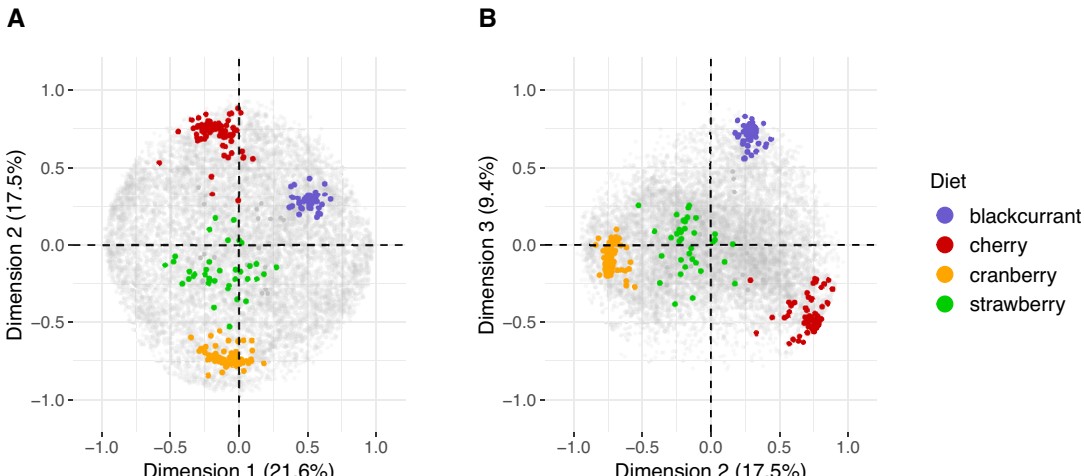

**Appendix 1—figure 6.** Plots of all individual ions of the principal component analysis of fruit metabolomes. (**A**) Dimension 1 vs. dimension 2, and (**B**) dimension 2 vs. dimension 3. Fruit-specific ions are indicated in color, all other ions are indicated in gray.

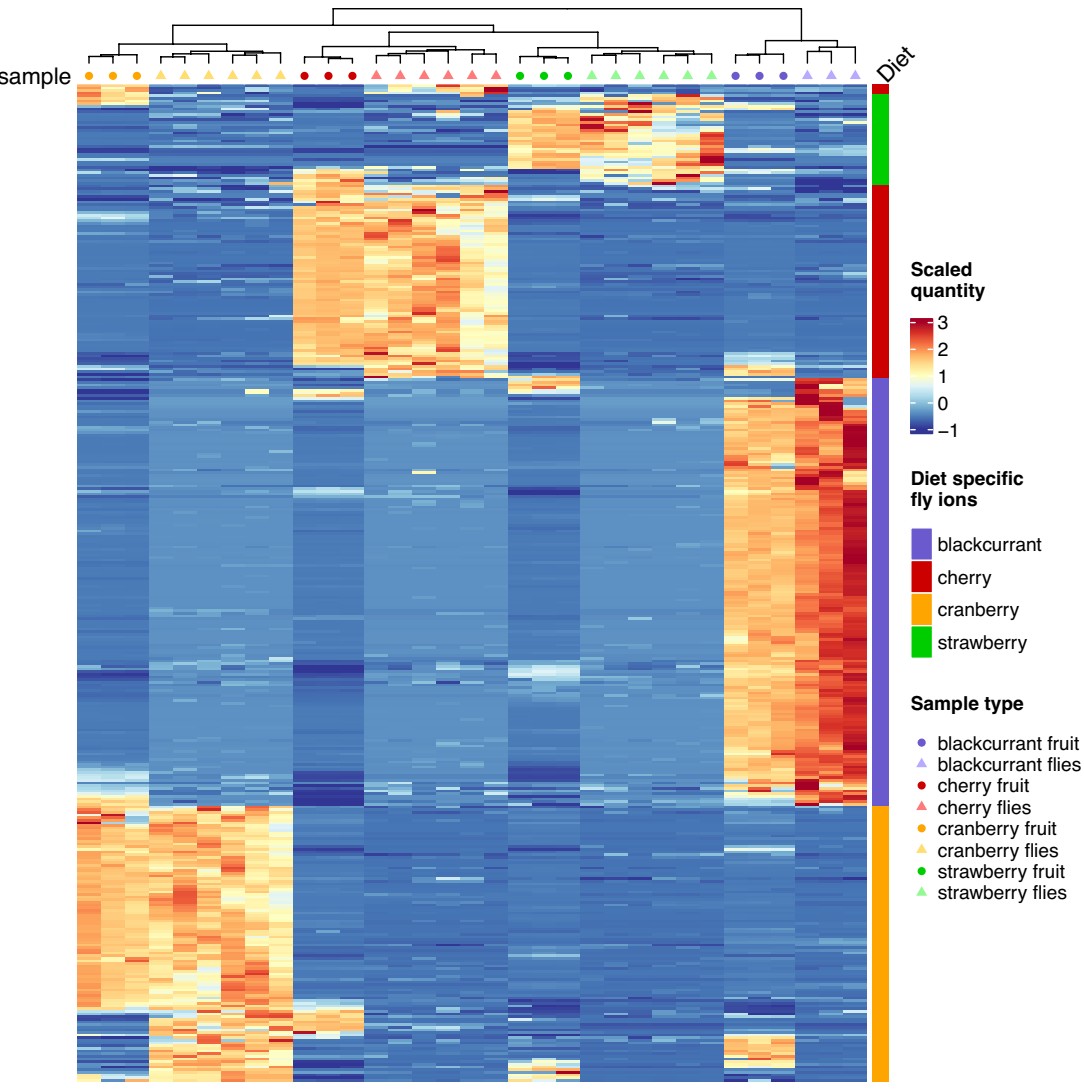

**Appendix 1—figure 7.** Classification of fruit and fly samples following selection of diet-specific ions through GLM with Elastic Net regularization using the raw *fly* dataset. Heatmap scale colors indicate the relative quantity of each ion (row) of the 'large' list of diet-specific ions (specificity indicated by discrete colors; see main text for details).

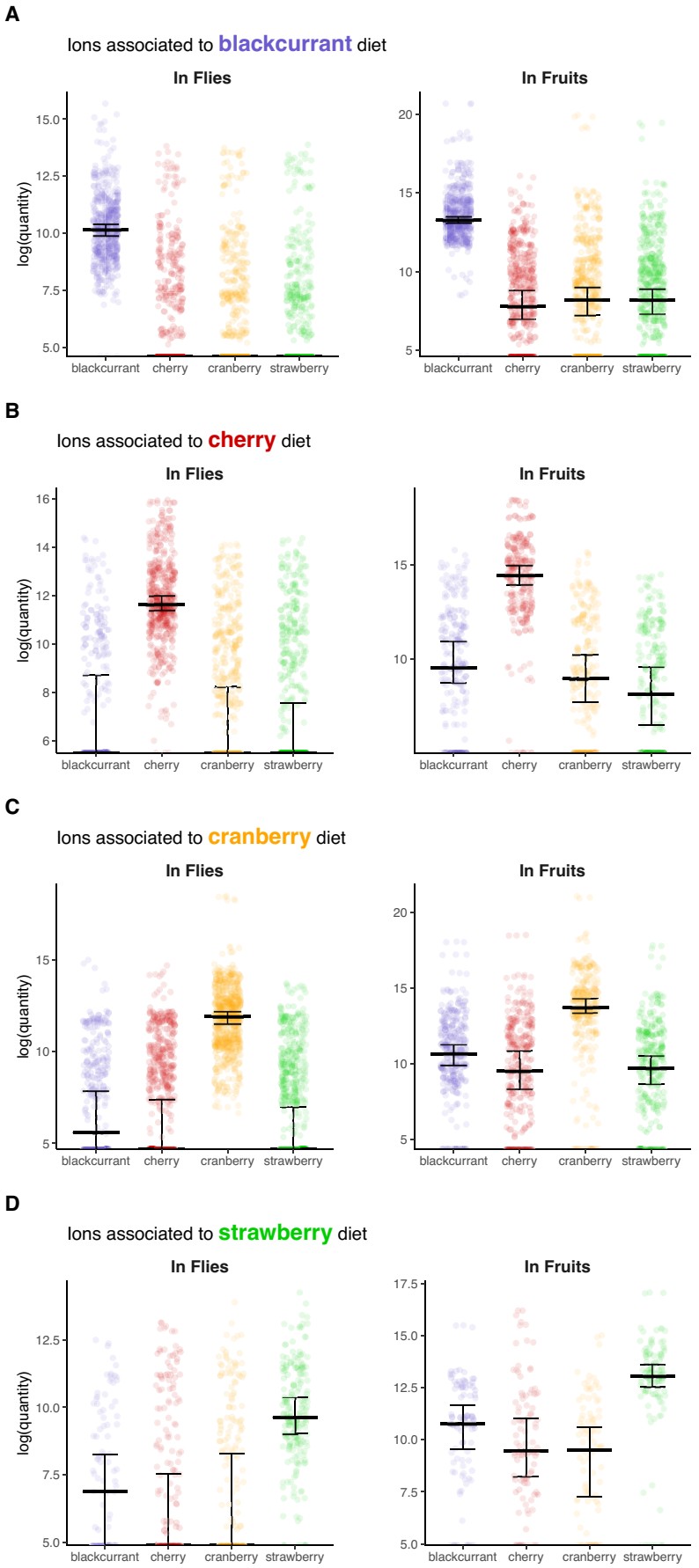

**Appendix 1—figure 8.** Quantitative levels of diet-specific fly ions in flies and fruits. (**A**) Blackcurrant, (**B**) Cherry, (**C**) Cranberry, (**D**) Strawberry. Medians and their confidence intervals are represented by black bars.

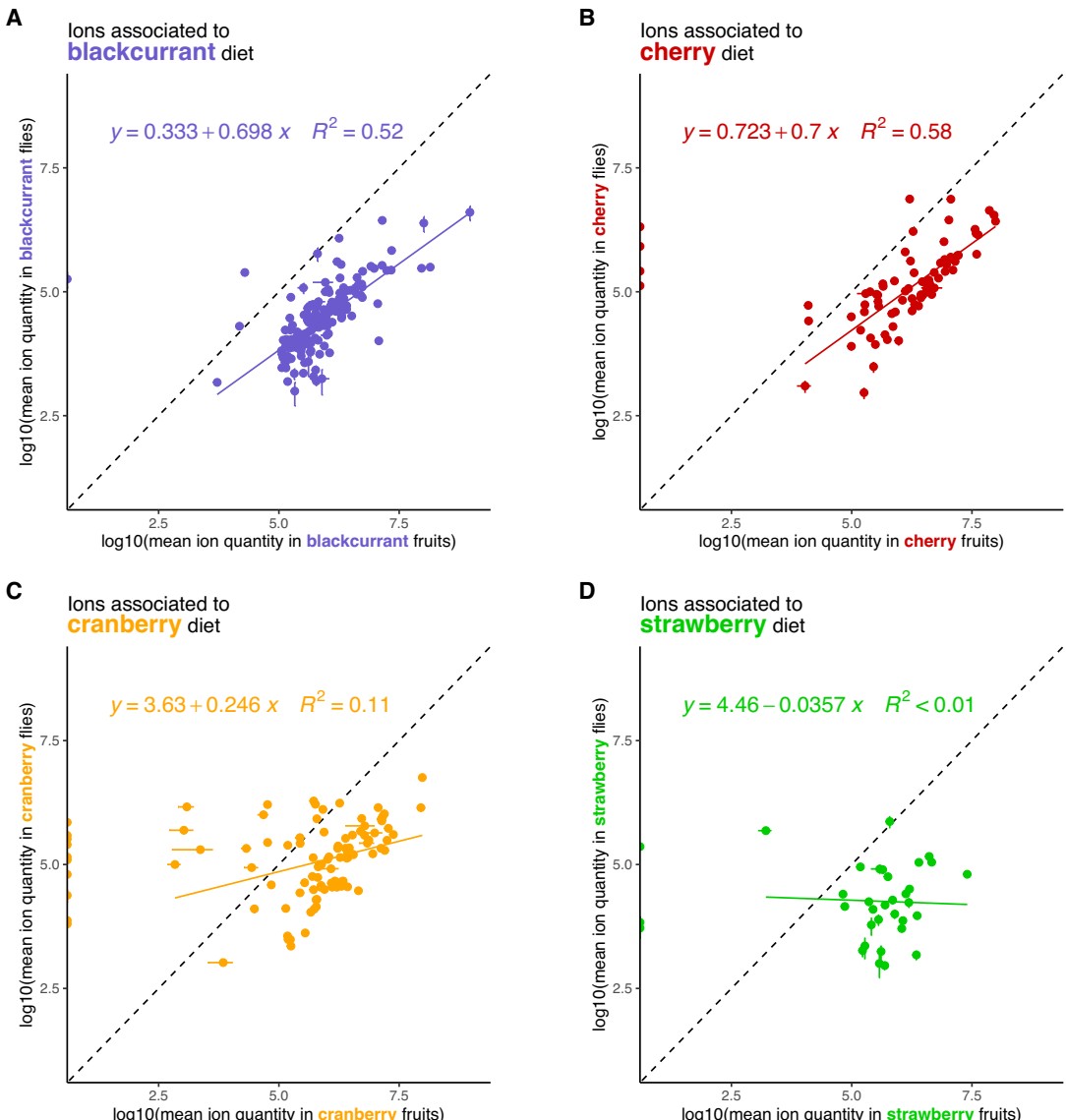

**Appendix 1—figure 9.** Relationships between the quantities of diet-specific fly ions in fruit and flies of the following fruit-fly pairs. (**A**) Blackcurrant, (**B**) Cherry, (**C**) Cranberry, (**D**) Strawberry.

## Conclusion

The analysis of the raw *fly* dataset yielded equivalent results to those using the *fly* dataset corrected for the presence of artificial medium-derived metabolites. This confirms that our conclusions are robust to the presence of these metabolites: metabolomes of flies fed on different fruits are mostly identical and 'metabolic generalism' (and not 'multi-host metabolic specialism') seems to be the mechanism promoting diet breadth.

For further comparison between results using both raw and controlled *fly* datasets, we present below a table grouping all analyses and their respective figures using both datasets.

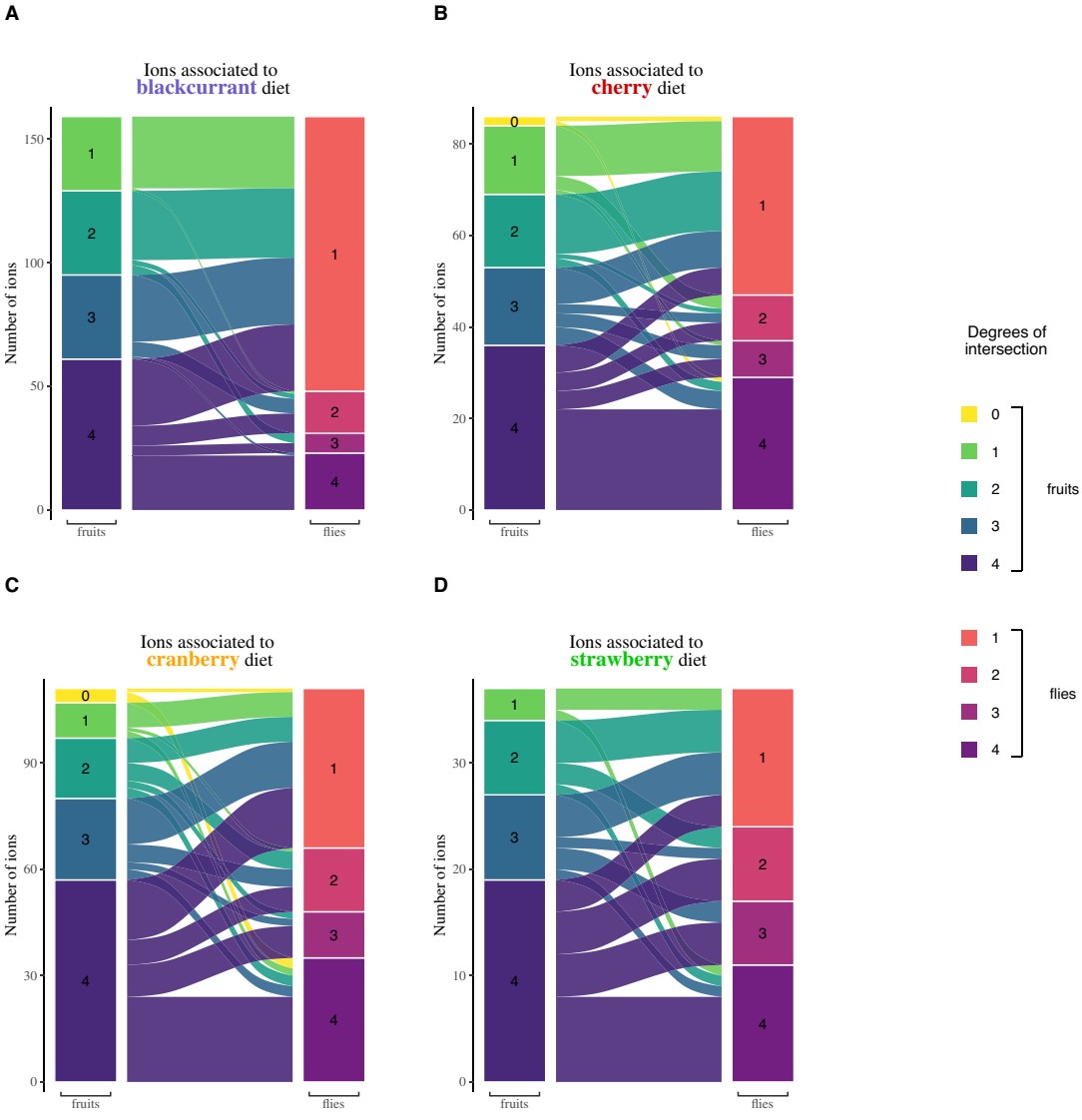

**Appendix 1—figure 10.** Intersection sizes (in fruits and flies) and relationships of fly ions specific to the following diets. (**A**) Blackcurrant, (**B**) Cherry, (**C**) Cranberry, (**D**) Strawberry. Ions that are shared within fruits or within flies show a higher degree of intersection.

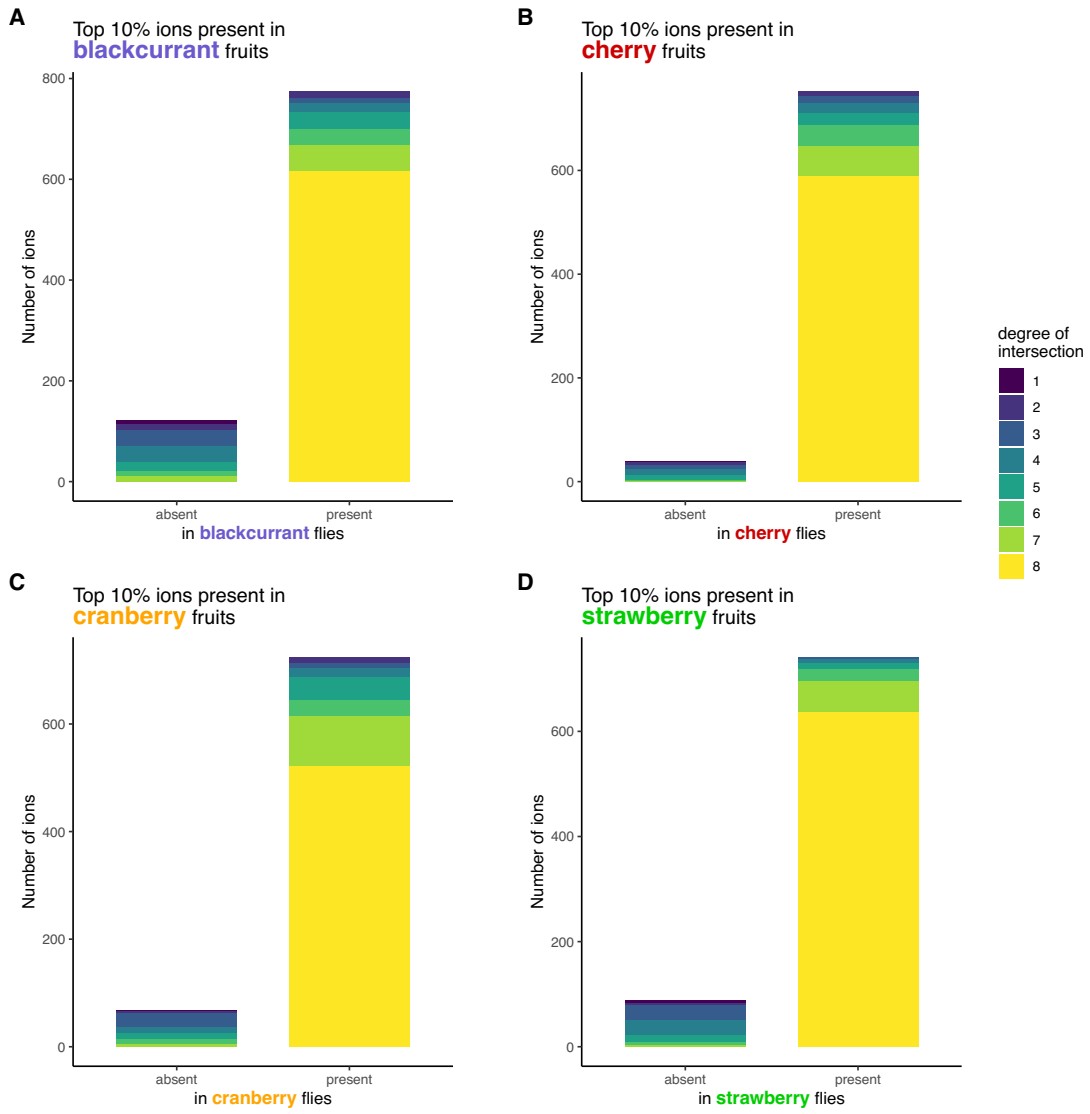

**Appendix 1—figure 11.** Number and degree of intersection of major fruit ions relative to their presence in the consumer flies. (**A**) Blackcurrant, (**B**) Cherry, (**C**) Cranberry, (**D**) Strawberry.

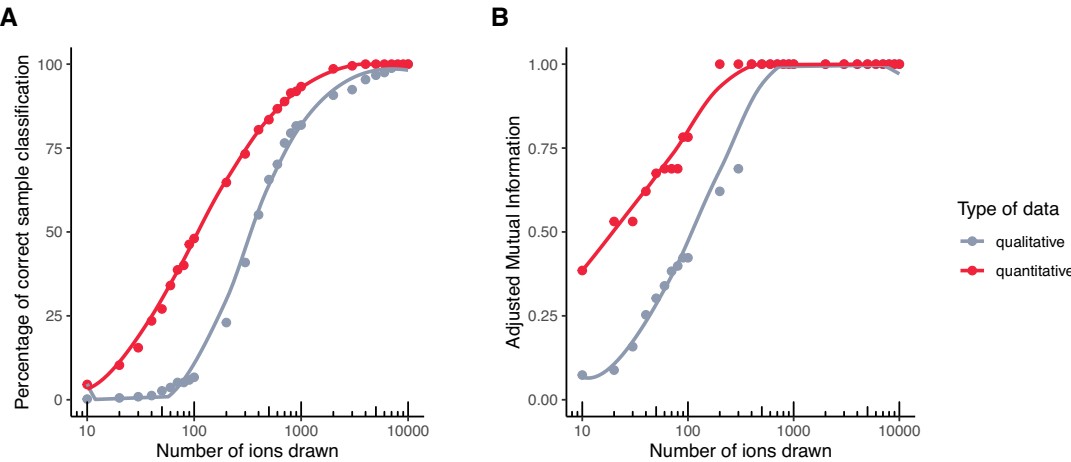

**Appendix 1—figure 12.** Classification performance of qualitative vs. quantitative fly datasets to infer diet. (**A**) Percentage of correct classification, and (**B**) Adjusted mutual information.

**Appendix 1—table 1.** Figure names for each analysis performed on raw and controlled fly dataset.

| Analysis | Raw fly dataset | Controlled fly dataset |
| --- | --- | --- |
| Relation between ion quantity in fruit vs. in flies that developed on the same fruit | *Appendix 1—figure 1* | *Figure 1—figure supplement 4* |
| Overall trends of shared ions between all fruits and all flies | *Appendix 1—figure 2* | *Figure 2* |
| Relationships between ions respectively found in fruits and flies, according to their degree of intersection | *Appendix 1—figure 3* | *Figure 3* |
| Fruits and flies' metabolomes Principal Component Analysis | *Appendix 1—figure 4* | *Figure 3—figure supplement 1* |
| Flies' metabolomes Principal Component Analysis | *Appendix 1—figure 5* | *Figure 4* |
| All individual ions of the Principal Component Analysis of fruit metabolomes | *Appendix 1—figure 6* | *Figure 4—figure supplement 1* |
| Classification of fruit and fly samples following selection of diet-specific ions through GLM with Elastic Net regularization | *Appendix 1—figure 7* | *Figure 4—figure supplement 2* |
| Quantitative levels of diet-specific fly ions in flies and fruits | *Appendix 1—figure 8* | *Figure 4—figure supplement 3* |
| Relationships between the quantities of diet-specific fly ions in fruit and flies of the following fruit-fly pairs | *Appendix 1—figure 9* | *Figure 4—figure supplement 4* |
| Intersection sizes (in fruits and flies) and relationships of fly ions specific to the following diets | *Appendix 1—figure 10* | *Figure 4—figure supplement 5* |
| Number and degree of intersection of major fruit ions relative to their presence in the consumer flies | *Appendix 1—figure 11* | *Figure 4—figure supplement 6* |
| Classification performance of qualitative vs. quantitative fly datasets to infer diet | *Appendix 1—figure 12* | *Figure 4—figure supplement 7* |

