## [Editor Report]

Most insect herbivores specialize in one or a few plants, but some species manage to be generalists and consume a wide variety of plant species. How generalists are able to cope with a variety of bioactive plant metabolites is not well understood. This important study uses untargeted metabolomics to identify patterns of metabolic generalism in herbivores. The study presents a convincing global chemical comparison of metabolites in insect diets with metabolites found in the insects, supporting inferences about the mechanisms underlying dietary generalism.

---

## [Decision Letter]

**Decision letter after peer review:**

Thank you for submitting your article "Metabolic consequences of various fruit-based diets in a generalist insect species" for consideration by *eLife*. Your article has been reviewed by 3 peer reviewers, one of whom is a member of our Board of Reviewing Editors, and the evaluation has been overseen by Meredith Schuman as the Senior Editor. The reviewers have opted to remain anonymous.

On the basis of metabolomics methods, this manuscript identified the background of the generalization of insect dietary preferences based on a novel hypothesis which the authors termed the "general metabolism hypothesis". Many studies in this area focus on how plants and specialized insect herbivores counter-adapt, and the authors present an interesting approach. The following concerns must be resolved through revision before being published in *eLife*. Please respond only to these essential revisions in your response letter and your revision, which are determined from the consultative review process. In doing so you may refer to relevant comments from the individual reviews.

1) In this experiment, fruit extract was added to an artificial diet rather than (additionally) using various fruits, which decreases its ecological relevance. Moreover, it is questioned whether artificial diet contamination affects the insect metabolome data. This problem can be addressed by analyzing direct fruit feeding in addition to the artificial diet data.

2) It is also essential to address the concerns on the analysis process for metabolomics data raised by reviewer #3.

3) Moreover, as mentioned by reviewer #1, the excessive focus on the model organism limits understanding the value of the manuscript. This study requires further discussion regarding scalability from the perspective of a generalist's diet breath.

*Reviewer #1 (Recommendations for the authors):*

To emphasize the implication and novelty of this study, the authors should revise the flow of the manuscript according to the main research hypotheses. Detailed suggestions are as follows:

1) Introduction

– Research questions (L86~L90) should be rewritten according to the two main research hypotheses.

– The last paragraph in the Introduction section contains the summarized results of the study (L93~L96). This part should be removed or moved to the Discussion section.

2) Results

– Description of the contamination and sequestration (accumulation) of fruit-specific metabolites in the fly is not clearly separated. More description about these two aspects is needed.

– The premise of the PCA of the integral dataset (L185~L187) seems inappropriate because of the fundamental difference between plant (fruit) and animal (fly). Moreover, the fly diet contained not only fruit puree. Even if the diets are the main influence on flies' metabolomes, the default constituents of the fly body might make bias in the multi-component analysis. From that point, in Figures 1F and 1G, the expected results from the hypotheses are also inappropriate when the whole dataset and ions are used. Authors should modify the design of the analysis in Figures 1F and 1G, and interpret the results carefully.

3) Discussion

– As mentioned in the Introduction part, the phylogenetic relationship of the studied berries [blackcurrant (Grossulariaceae), cherry and strawberry (Rosaceae), and cranberry (Ericaceae)] should be discussed to assess the phylogenetic diet breadth. Metabolomic profiles as in PCA results do not correspond to the phylogenetic relationship of the four studied species.

– Probable mechanism of the diet generalism of studied species is described well. On the other hand, the overall readability could be enhanced by rearrangement of the paragraphs according to the description of the results or design of Figure 1.

– The last two paragraphs seemed to be far from the main research context. If the authors intended to discuss the plant domestication and metabolism of fly, developed phenotypes during domestication about all studied species should be more discussed.

4) Conclusion

– Overall sentences of the Conclusion section do not correspond to the main context. Except for the first sentence, sentences seem to be appropriate to be Introduction section. Authors should re-write the conclusion from the key finding and implications of the research.

*Reviewer #3 (Recommendations for the authors):*

1. Comparing the diet metabolome and frass metabolome of D. suzukii would provide cleaner results without interference from the insect metabolites. On the same line, a proper control metabolome of D. suzukii fed on an artificial diet without fruits shall be included.

2. The authors simply merged positive mode data and negative mode data, however, many metabolites could be detected in both ionization modes, this redundancy would influence the number of shared ions, and thus, shall be further cleaned up before analysis.

3. In P8, line 121, the authors claimed "qualitative data alone enabled perfect discrimination between fruit", while in Figure 1—figure supplement 2, it seems that the great majority of ions (4996) are indiscriminate and shared among 4 fruits.

4. In P13, line 252, the authors stated that "A survey of the published literature showed that many of the metabolites identified here in specific fruits (and flies that were grown on them) had already been detected and identified in the same fruits (Figure 5 & Supplementary Table 1), thereby supporting their correct identification." I don't see these are evidence of correct identification, at least an MS/MS comparison shall be provided to support these annotations.

[Editors' note: further revisions were suggested prior to acceptance, as described below.]

Thank you for resubmitting your work entitled "Metabolic consequences of various fruit-based diets in a generalist insect species" for further consideration by *eLife*. Your revised article has been evaluated by Meredith Schuman (Senior Editor) and a Reviewing Editor.

This important study uses untargeted metabolomics to help us understand how some herbivores are able to be generalists, rather than specializing in the metabolism of specific plant species. This is an important area, since little is known about how generalist insect species metabolize their food. Particularly, this study provides a valuable global chemical comparison of how diverse diet metabolites are processed by a generalist insect species. The evidence, potentially solid, is currently incomplete: see the remaining comments below.

The majority of concerns raised by reviewers were addressed effectively. In spite of this, the method section is too simplified and does not provide enough information to readers, as well as the fact that artificial diet control analysis and results were not provided – the supplementary information should at least include this information.

---

## [Author Response]

On the basis of metabolomics methods, this manuscript identified the background of the generalization of insect dietary preferences based on a novel hypothesis which the authors termed the "general metabolism hypothesis". Many studies in this area focus on how plants and specialized insect herbivores counter-adapt, and the authors present an interesting approach. The following concerns must be resolved through revision before being published in eLife. Please respond only to these essential revisions in your response letter and your revision, which are determined from the consultative review process. In doing so you may refer to relevant comments from the individual reviews.

We thank the Editor and Reviewers for their consideration of our manuscript. We agree that the main originality of our work is the joint analysis of plants’ and herbivores’ chemistries and we tried to improve our manuscript according to the referees’ suggestions. Please find our detailed responses to all comments below (lines numbers correspond to the manuscript with highlighted changes file).

1) In this experiment, fruit extract was added to an artificial diet rather than (additionally) using various fruits, which decreases its ecological relevance.

We agree with the Editor and Reviewers that using natural fruits in our experiment would have increased the ecological relevance of our study. However, previous experiments by our group showed that artificial fruit media provide a relatively accurate sensorial representation of the actual fruits individuals encounter in the field. Indeed, Olazcuaga et al. 2022 found that female oviposition preference and larval performance was higher on artificial fruit medium corresponding to the fruit from which the individuals originated than on fruit media corresponding to other fruits.

In addition, we chose to use fruit purees supplemented with a baseline artificial diet for the following reasons:

1. In many dipteran species and *Drosophila* species in particular, fruits alone do not provide all the chemical compounds necessary for larval development. Specifically, proteins are mainly harvested from microorganisms. While natural fruits would have provided such elements to our larval and adult flies, it would have been impossible to control for a homogenous input of proteins between fruits and among replicates. Besides protein input, natural microorganisms would probably alter other aspects of fruit composition, together with consumers’ microbiota. Such uncontrolled cascading effects would be difficult to account for, and reaching a conclusion about our main question could have been impossible.

2. We used fruit purees that were as close as possible to natural fruits: while they were industrially produced in order to allow for a homogeneous composition between replicates, all fruit purees were organic, and the transformation steps only included mechanical transformation and pasteurization. In particular, no sugars, coloring, or preservatives were added to fruit purees.

3. While an artificial diet basis was included in our rearing media, its share was kept to a minimum, since our recipe is 60% of fruit puree and 40% of the artificial diet. Our artificial media are thus far from the standard media used in regular laboratory rearing procedures.

4. The composition of the artificial part of our fruit media was relatively simple, consisting mainly of yeast-based protein sources.

We thus agree that the ecological relevance of our study would have improved by using of natural fruit. However, to reach firm conclusions about the metabolic processes underpinning generalist diets in insects, we needed to maintain controlled dietary inputs throughout the insects’ development. We consider that our choice of an artificial medium consisting largely of natural fruits is the preferable balance between controlled experimental procedures and ecological relevance. We now include modifications l479-487 to better explain this point.

We also wish to note that while experimental evaluation of the interaction between bacterial communities, fruits, and their consumers was beyond the scope of our present study, the addition of this ecologically relevant layer of complexity constitutes a captivating avenue for future research. We could add this perspective to our manuscript if the Editor/Reviewers consider it noteworthy.

Moreover, it is questioned whether artificial diet contamination affects the insect metabolome data. This problem can be addressed by analyzing direct fruit feeding in addition to the artificial diet data.

Because some chemical inputs are required from the artificial media (e.g., proteins) to complete insects’ development, it is true that this artificial diet component is included in the insects’ metabolome.

However, using an artificial diet component with industrial yeast enabled us to certify that all fruits and replicates are homogeneous regarding these elements. Therefore, the only variation in the diets of all insects was the fruit component, with all other elements being equal and consistent. This allowed a direct comparison between diets and their effect on insects’ metabolomes.

As stated before, we think that direct fruit feeding could either:

mix relative contributions of microorganisms and fruit content to the insects’ metabolomes, with unknown variance between replicates, if fruits are not controlled for their bacterial content,fail to complete the insects’ development if fruits are rid of their bacterial content.

We thus feel that direct fruit feeding may not be the appropriate procedure to exclude the effect of the artificial component of the diet from the insects’ metabolomes. Additional layers of biological complexity could add much noise to the experimental data and preclude any conclusion if not considered/controlled for.

Another way to control whether the artificial diet could alter our conclusion regarding the effect of fruit on the insect metabolome is by including insects that have only experienced the artificial diet. As stated in more details below, we did include such controls in our original experiment, at each generation. The inclusion of such control individuals did not alter our conclusions (see below). We modified our manuscript to include this result (l.110-113 in the Results section and l.584-589 in the Methods section), which will certainly strengthen the reader’s confidence about the relevance of the study.

2) It is also essential to address the concerns on the analysis process for metabolomics data raised by reviewer #3.

The first concern of Reviewer #3 on the analysis process regards our ability to interpret patterns when “fruit-derived” metabolites and “insect-derived” metabolites are mixed together. We now explain that proper fly controls were processed and illustrate below that their inclusion in the dataset results in similar patterns and identical conclusions. Please find a detailed answer in the Reviewer #3 section.

The second concern of Reviewer #3 regards our methodology for peak selection. We now detail our procedure and explain our choice of threshold in the Reviewer #3 section.

The third concern regards our methodology for peak annotation. We are now providing MS2 data as recommended by Reviewer #3 and present our annotation results in full detail in Supplementary Table 1. Please find a detailed answer in the Reviewer #3 section.

3) Moreover, as mentioned by reviewer #1, the excessive focus on the model organism limits understanding the value of the manuscript. This study requires further discussion regarding scalability from the perspective of a generalist's diet breath.

We agree with the Editor/Reviewers on this important comment. We thus modified our Discussion accordingly and took care to use terms for which each point could be made at the correct scale (most of which are herbivores or frugivores, see Discussion). The peculiarity of our model system is now only discussed in a single occasion and serves as an example (l.387-396). We feel that our Discussion section and its biological value has been significantly enhanced by this comment.

Reviewer #1 (Recommendations for the authors):To emphasize the implication and novelty of this study, the authors should revise the flow of the manuscript according to the main research hypotheses. Detailed suggestions are as follows:1) Introduction– Research questions (L86~L90) should be rewritten according to the two main research hypotheses.

We have rewritten the (now) last paragraph of the introduction section accordingly (l.86-93).

– The last paragraph in the Introduction section contains the summarized results of the study (L93~L96). This part should be removed or moved to the Discussion section.

The previously last paragraph of the Introduction section has now been removed, according to the comment. We modified it to be able to move it in discussion.

2) Results– Description of the contamination and sequestration (accumulation) of fruit-specific metabolites in the fly is not clearly separated. More description about these two aspects is needed.

We have extended the corresponding paragraph according to the Reviewers’ comment (l.136-146).

– The premise of the PCA of the integral dataset (L185~L187) seems inappropriate because of the fundamental difference between plant (fruit) and animal (fly). Moreover, the fly diet contained not only fruit puree. Even if the diets are the main influence on flies' metabolomes, the default constituents of the fly body might make bias in the multi-component analysis. From that point, in Figures 1F and 1G, the expected results from the hypotheses are also inappropriate when the whole dataset and ions are used. Authors should modify the design of the analysis in Figures 1F and 1G, and interpret the results carefully.

It may be the only point where we slightly disagree with Reviewer #1 on a minor aspect of this question, but still agree with her/his overall comment and performed the desired modifications.

From the point of view of the investigative technique of LC-MS, there is no fundamental difference between plant and animal metabolites. The methodology used here is totally blind to the nature of the processed samples (plant, animal or others), and simply recapitulates their chemical composition, in a standard way.

We agree with Reviewer #1 that fruits are not the only component in the fruit media and have addressed this point in detail below (see answers to Reviewer #3). Briefly, we now show the Editor & Reviewers that accounting for the artificial medium component of flies’ diet does not modify our results.

In particular, regarding Figure 1F & 1G, we show below that controlling for the artificial medium component of the flies’ diet has little impact on our results: while differently fed flies are slightly more dispersed, they still constitute a single cluster, and conclusions drawn from this analysis remain. With this controlled data, we show that the observed clustering of flies is not derived from the addition of a small proportion of artificial medium to their diet, but is derived from metabolites that originate directly or indirectly from fruits.

We agree however that even if fruits are the main influence on flies’ metabolome, a PCA could display a much less caricatural view than illustrated in Figure 1G. We have thus modified Figure 1G to better represent what could be the case of ‘multi-host metabolic specialism’: in this case, basic fly metabolomes would still be apart from fruit metabolomes (i.e., separated on dimension 1) but the diversity of metabolomic responses to various diets would align them on their fruit on the second dimension (e.g., because of high levels of diet specific metabolites being produced). In the case of a single metabolomic response to all diets (i.e., ‘metabolic generalism’), flies would also cluster on the second dimension of the PCA (e.g., because of the production of common metabolites irrespective of diet). We modified the Figure 1 caption accordingly (l.1028-1032). We thank Reviewer #1 for this comment, which has improved the presentation of our hypotheses.

3) Discussion– As mentioned in the Introduction part, the phylogenetic relationship of the studied berries [blackcurrant (Grossulariaceae), cherry and strawberry (Rosaceae), and cranberry (Ericaceae)] should be discussed to assess the phylogenetic diet breadth. Metabolomic profiles as in PCA results do not correspond to the phylogenetic relationship of the four studied species.

We would like to thank Reviewer #1 for this very interesting comment that we did not previously develop. Indeed, the biochemical signal between fruits does not match the phylogenetic signal of fruit species. We agree with Reviewer #1 that this observation has ample consequences when trying to study generalism, as taxonomy may not represent a good scale to measure adaptive challenges. We now have made this clearer in the revised version of the manuscript (i) by adding a new figure to better illustrate this observation (Figure 1—figure supplement 3), and (ii) by discussing this point as recommended (l.131-135 and 338-343).

– Probable mechanism of the diet generalism of studied species is described well. On the other hand, the overall readability could be enhanced by rearrangement of the paragraphs according to the description of the results or design of Figure 1.

We thank Reviewer #1 for their comment on our description of diet generalism. However, the discussion is now modified to address the editor's main comment: to be less focused on our specific *D. suzukii* results. Briefly, our Discussion section now articulates the following subjects that are of interest to a broad readership: diversity and taxonomy of fruit composition, generality of qualitative response and plasticity in quantitative response to various diets, impact on the understanding of paradoxical results in diet generalism literature, possible origin of diet generalism, possible costs and benefits of metabolic generalism, relation between insect generalism and plant domestication, technical possibility of diet tracing in the wild and a few perspectives for future research. We feel that this discussion already sweeps a wide spectrum of general issues in diet generalism and do not wish to overburden it.

We however tried to broaden the scope of all Discussion paragraphs by limiting the reference to the specific metabolism of our model species (only l.386-396 remaining), as well as including more references from plant and fruit biochemistry studies. In particular, we now cite the works of Whitehead et al. to better link our results with scientific communities working on the evolution of plant biochemistry and frugivores’ diet breadth.

– The last two paragraphs seemed to be far from the main research context. If the authors intended to discuss the plant domestication and metabolism of fly, developed phenotypes during domestication about all studied species should be more discussed.

We agree with Reviewer #1 that the previous last two paragraphs are less related to the main research question of diet generalism. We chose to maintain those paragraphs in the current version for the following reasons: (i) the paragraph of diet tracing is a small yet important contribution to the possible study of diet generalism in the wild using untargeted metabolomics, and (ii) we developed our point about plant domestication and generalism in the current version.

Briefly, our main point about plant domestication and a possible impact on herbivore generalism is that the biochemical features most seeked by human domesticators may trade-off with plant defenses. This was demonstrated by a meta-analysis by Whitehead et al. (2017) that we now cite, and while this may not be true for all plant tissue, this was especially the case in fruits, and generally for harvested organs in crops. In turn, these biochemical changes are expected to lead to (i) increased generalism in frugivores and (ii) increased presence of non-lethal plant secondary metabolites in frugivores. We modified the corresponding Discussion paragraph accordingly (l.417-422).

4) Conclusion– Overall sentences of the Conclusion section do not correspond to the main context. Except for the first sentence, sentences seem to be appropriate to be Introduction section. Authors should re-write the conclusion from the key finding and implications of the research.write the conclusion from the key finding and implications of the research.

>> We agree that our previous Conclusions were rather perspectives and shortened and moved these sentences at the end of our Discussion section. We now provide a Conclusion that highlights how our study contributes to a deeper understanding of dietary generalism (l.454-459).

Reviewer #3 (Recommendations for the authors):1. Comparing the diet metabolome and frass metabolome of D. suzukii would provide cleaner results without interference from the insect metabolites. On the same line, a proper control metabolome of D. suzukii fed on an artificial diet without fruits shall be included.

We agree with Reviewer #3 that frass metabolome would be an interesting addition to the data to better infer the excretion part of the metabolism of an herbivorous insect to a variety of diets. By observing distinctive fruit components that are accumulated, we here start to comprehend the metabolization process of various diets. We however feel that to avoid confounding effects, frass metabolome data would be better complemented with microbiota data, or even better, microbiota experimental control. This was beyond our primary objective to bring a large-spectrum biochemistry point of view to the community and would lead to a significantly larger amount of work. We feel that the venue underlined by Reviewer #3 is very promising for the future.

Regarding flies fed on an artificial diet without fruit, we did such proper controls (see above) and the inclusion of such controls did not modify our conclusions. We hope that the presentation of these results and the modification in the Results and Material and Methods sections (l.110-113; l.512-515, and l.583-589) will convince the Reviewer and all readers that our results do not depend on the presence of a fraction of artificial medium in the flies’ diet.

2. The authors simply merged positive mode data and negative mode data, however, many metabolites could be detected in both ionization modes, this redundancy would influence the number of shared ions, and thus, shall be further cleaned up before analysis.

We agree with Reviewer #3 that some metabolites could be detected in both ionization modes.

We thus tested whether restricting our analyses to a given ionization mode, positive or negative, would alter our results and conclusions. In Author response images 1-3 we show the Figures 2, 3 and 4 redrawn using either positive or negative modes. None of our results is significantly altered (same patterns of common and uncommon ions, same ability to discriminate flies based on their diet), so merging positive and negative datasets does not incur an important bias to our analyses and was kept. We now indicate that our analyses are robust to ionization mode in our manuscript (l.577-579).

**Author response image 1. sa2fig1:** Left = negative ions; right = positive ions.

**Author response image 2. sa2fig2:** Left = negative ions; right = positive ions.

**Author response image 3. sa2fig3:** Above = negative ions; below = positive ions.

3. In P8, line 121, the authors claimed "qualitative data alone enabled perfect discrimination between fruit", while in Figure 1—figure supplement 2, it seems that the great majority of ions (4996) are indiscriminate and shared among 4 fruits.

We agree with Reviewer #3 that the great majority of ions are shared among the four fruits. However, it is also possible to distinguish fruits based on the qualitative data alone. As indicated in our sentence and our Figure, several hundred ions are diagnostic of each fruit (from at least 287 to 650 private ions depending on the fruit). We modified our sentence to reflect that fruits also share a large part of their ions (l.127).

4. In P13, line 252, the authors stated that "A survey of the published literature showed that many of the metabolites identified here in specific fruits (and flies that were grown on them) had already been detected and identified in the same fruits (Figure 5 & Supplementary Table 1), thereby supporting their correct identification." I don't see these are evidence of correct identification, at least an MS/MS comparison shall be provided to support these annotations.

As stated previously, Supplementary Table 1 now details all information used for metabolite annotation: validation standard for 30 of 71 identified compounds, and MS/MS data comparison with mass spectral databases as well as with published literature. We now also provide an identification confidence level for each compound, as proposed by Schymanski et al. 2014 (full reference included in the metadata of Supplementary Table 1).

[Editors' note: further revisions were suggested prior to acceptance, as described below.]

This important study uses untargeted metabolomics to help us understand how some herbivores are able to be generalists, rather than specializing in the metabolism of specific plant species. This is an important area, since little is known about how generalist insect species metabolize their food. Particularly, this study provides a valuable global chemical comparison of how diverse diet metabolites are processed by a generalist insect species. The evidence, potentially solid, is currently incomplete: see the remaining comments below.The majority of concerns raised by reviewers were addressed effectively. In spite of this, the method section is too simplified and does not provide enough information to readers, as well as the fact that artificial diet control analysis and results were not provided – the supplementary information should at least include this information.

We thank the Associate Editor and the Reviewing Editor for their comments. We fully understand the absolute necessity of providing the readers all information for correct assessment and replication of our study. This is especially true for the artificial diet control analysis and results, that are crucial to our study. That is why we now provide the artificial diet control analysis in full and in the main text, together with the raw data analysis in full in an Appendix. We have accordingly redrawn all figures of the main text to correspond to the artificial diet control analysis (all previous figures from the raw data analysis are included in the Appendix). We thus hope that the full inclusion of both analyses will enable the reader to correctly assess our evidence regarding the validation of our hypotheses.

We also tackled the first point mentioned by providing more information in our methods section. We now provide additional information regarding our food media preparation (l. l.447-470), origin of stock flies (l.481-488, 494-496), artificial diet control procedure (l.509-513; 587-597), processing of metabolomic data (l.562-567; 572-573) and host use analysis (l.634-642). All changes are highlighted in the corresponding docx file, and our current methods section is now 3749 words long. If Reviewing editors still think there is major information missing for this section, please do not hesitate to be specific in your demand, as we might have missed an important expectation.

Since previous comments asked specifically about the treatment of metabolomic data under XCMS, along with providing along the necessary procedure and the XCMS-processed datasets, we also now provide the raw metabolomic dataset before any XCMS treatment, on the dedicated *Metabolights* repository. While our data files are deposited and our accession number was edited and included in our manuscript, please be aware that validation and public disclosure usually occur after at least a week. Please also note that all code and datasets (both raw and artificial diet controlled) were updated in the dataverse repository and in the Shiny visualization application mentioned in our manuscript.